# Technical note: A high-resolution inverse modelling technique for estimating surface CO₂ fluxes based on the NIES-TM - FLEXPART coupled transport model and its adjoint.

Shamil Maksyutov[1], Tomohiro Oda[2], Makoto Saito[1], Rajesh Janardanan[1], Dmitry Belikov[1,3], Johannes W. Kaiser[4], Ruslan Zhuravlev[5], Alexander Ganshin[5], Vinu K. Valsala[6], Arlyn Andrews[7], Lukasz Chmura[8], Edward Dlugokencky[7], László Haszpra[9], Ray L. Langenfelds[10], Toshinobu Machida[1], Takakiyo Nakazawa[11], Michel Ramonet[12], Colm Sweeney[7], Douglas Worthy[13]

[1]National Institute for Environmental Studies, Tsukuba, Japan
[2]NASA Goddard Space Flight Center, Greenbelt, MD, USA/Universities Space Research Association, Columbia, MD, USA
[3]now at Chiba University, Chiba, Japan
[4]Deutscher Wetterdienst, Offenbach, Germany
[5]Central Aerological Observatory, Dolgoprudny, Russia
[6]Indian Institute for Tropical Meteorology, Pune, India
[7]Earth System Research Laboratory, NOAA, Boulder, CO, USA
[8]AGH University of Science and Technology, Krakow, Poland
[9]Research Centre for Astronomy and Earth Sciences, Sopron, Hungary
[10]Climate Science Centre, CSIRO Oceans and Atmosphere, Aspendale, VIC, Australia
[11]Tohoku University, Sendai, Japan
[12]Laboratoire des Sciences du Climat et de l'Environnement, LSCE-IPSL, Gif-sur-Yvette, France
[13]Environment and Climate Change Canada, Toronto, Canada

*Correspondence to*: S. Maksyutov (shamil@nies.go.jp)

## Abstract

We developed a high-resolution surface flux inversion system based on the global Lagrangian-Eulerian coupled tracer transport model composed of the National Institute for Environmental Studies Transport Model (NIES-TM) and the FLEXible PARTicle dispersion model (FLEXPART). The inversion system is named NTFVAR (NIES-TM-FLEXPART-variational) as it applies a variational optimization to estimate surface fluxes. We tested the system by estimating optimized corrections to natural surface CO₂ fluxes to achieve the best fit to atmospheric CO₂ data collected by the global in-situ network, as a necessary step towards the capability of estimating anthropogenic CO₂ emissions. We employed the Lagrangian particle dispersion model (LPDM) FLEXPART to calculate surface flux footprints of CO₂ observations at a

spatial resolution of $0.1° \times 0.1°$. The LPDM is coupled with a global atmospheric tracer transport model (NIES-TM). Our inversion technique uses an adjoint of the coupled transport model in an iterative optimization procedure. The flux error covariance operator was implemented via implicit diffusion. Biweekly flux corrections to prior flux fields were estimated for the years 2010-2012 from in-situ $CO_2$ data included in the Observation Package (ObsPack) dataset. High-resolution prior flux fields were prepared using the Open-Data Inventory for Anthropogenic Carbon dioxide (ODIAC) for fossil fuel combustion, the Global Fire Assimilation System (GFAS) for biomass burning, the Vegetation Integrative SImulator for Trace gases (VISIT) model for terrestrial biosphere exchange, and the Ocean Tracer Transport Model (OTTM) for oceanic exchange. The terrestrial biospheric flux field was constructed using a vegetation mosaic map and a separate simulation of $CO_2$ fluxes at a daily time step by the VISIT model for each vegetation type. The prior flux uncertainty for the terrestrial biosphere was scaled proportionally to the monthly mean Gross Primary Production (GPP) by the Moderate Resolution Imaging Spectroradiometer (MODIS) MOD17 product. The inverse system calculates flux corrections to the prior fluxes in the form of a relatively smooth field multiplied by high-resolution patterns of the prior flux uncertainties for land and ocean, following the coastlines and fine-scale vegetation productivity gradients. The resulting flux estimates improved the fit to the observations taken at continuous observation sites, reproducing both the seasonal and short-term concentration variabilities, including high $CO_2$ concentration events associated with anthropogenic emissions. The use of a high-resolution atmospheric transport in global $CO_2$ flux inversions has the advantage of better resolving the transported mixed signals from the anthropogenic and biospheric sources in densely populated continental regions. Thus it has a potential for achieving better separation between fluxes from terrestrial ecosystems and strong localized sources, such as anthropogenic emissions and forest fires. Further improvements in the modelling system are needed as our posterior fit was better than that of the National Oceanic and Atmospheric Administration (NOAA)'s CarbonTracker only for a fraction of the monitoring sites, mostly at coastal and island locations where background and local flux signals are mixed.

## 1 Introduction

Inverse modelling of the surface fluxes is implemented by using chemical transport model simulations to match atmospheric observations of greenhouse gases (GHGs). $CO_2$ flux inversions studies started from addressing large scale flux distributions (Enting and Mansbridge, 1989; Tans et al., 1990; Gurney et al., 2002; Peylin et al., 2013 and others) using background monitoring site data and global transport models at low and medium spatial resolutions, targeting extraction of the information on large and highly variable fluxes of $CO_2$ from terrestrial ecosystems and oceans. Merits of improving the spatial resolutions of global transport simulations to 9-25 km, which this study aims to demonstrate, have been also discussed by previous studies, such as Agusti-Panareda et al. (2019) and Maksyutov et al. (2008). However, global inverse modelling studies have never been conducted at these spatial resolutions. On the other hand, regional-scale

fluxes, such as national emissions of non-$CO_2$ GHGs, have been estimated using inverse modelling tools that rely on regional (mostly Lagrangian) transport algorithms which are capable of resolving surface flux contributions to atmospheric concentrations at resolutions from 1 to 100 km (Vermeulen et al., 1999; Manning et al., 2011; Stohl et al., 2009; Rodenbeck et al., 2009; Henne et al., 2016; He et al., 2018; Schuh et al., 2013; Lauvaux et al., 2016 and others). Extension of the regional Lagrangian inverse modelling to the global scale based on the combination of three-dimensional (3-D) global Eulerian model and Lagrangian model has been implemented in several studies (Rugby et al., 2011; Zhuravlev et al., 2013; Shirai et al., 2017). They have demonstrated an enhanced capability of resolving the regional and local concentration variabilities driven by fine-scale surface emission patterns, while the inverse modelling schemes rely on regional and global basis functions that yield concentration responses of regional fluxes at observational sites. A disadvantage of using regional basis functions in inverse modelling is the flux aggregation errors, as noted by Kaminski et al. (2001). This has been addressed by developing grid-based inversion schemes based on variational assimilation algorithms that yield flux corrections that are not tied to aggregated flux regions (Rodenbeck et al., 2003; Chevallier et al., 2005; Baker et al., 2006, and others). In order to implement a grid-based inversion scheme that is suitable for optimizing surface fluxes using a high-resolution atmospheric transport capability of the Lagrangian model, an adjoint of a coupled Eulerian-Lagrangian model is needed, as reported by Belikov et al. (2016).

In this study, we applied an adjoint of the coupled Eulerian-Lagrangian transport model, which is a revised version of Belikov et al. (2016), to the problem of surface flux inversion based on a coupled transport model with a spatial resolution of the Lagrangian model at 0.1° longitude-latitude. While global higher resolution transport runs can be implemented with coupled Eulerian-Lagrangian models (e. g. Ganshin et al., 2012), the choice of the model resolution in our inversion system was dictated mostly by the availability of the prior surface $CO_2$ fluxes.

The practical need for running high-resolution atmospheric transport simulations at the global scale is currently driven by expanding GHG observing capabilities towards quantifying anthropogenic emissions by observing GHGs at the vicinity of emission sources (Nassar et al., 2017; Lauvaux et al., 2020). These include observations in both background and urban sites, with tall towers, commercial airplanes, and satellites. At the same time, the focus of inverse modelling is evolving towards studies of the anthropogenic emissions, with a target of making better estimates of regional and national emissions in support of national and regional GHG emission reporting and control measures (Manning et al., 2011; Henne et al., 2016; Lauvaux et al., 2020). In that context, global-scale high-resolution inverse modelling approaches have the advantage in closing global budgets, while regional and national scale inverse modelling approaches with limited area models require boundary conditions normally provided by global model simulations with optimized fluxes. Often there is an additional degree of freedom introduced by allowing corrections to the boundary concentration distribution to improve a fit at the observation sites (Manning et al., 2011). As a result, the global total of regional emission estimates does not necessarily

match the balance constrained by global mean concentration trends. A global coupled Eulerian-Lagrangian model, such as Ganshin et al., (2012), has the potential for addressing both the objectives, that is closing the global balance and operating at the range of scales from a single city (Lauvaux et al., 2016) to a large country or continental scale. Here we report the development of a high-resolution inverse modelling technique that is suitable for application at a broad range of spatial scales. We applied it to the problem of estimating the distribution of $CO_2$ fluxes over the globe that provides the best model fit to the observations. In separate studies, the same inversion system was applied to inverse modelling of methane emissions (Wang et al., 2019; Janardanan et al., 2020).

The objective of this study is to optimize the natural $CO_2$ fluxes in order to provide background $CO_2$ concentration fields for estimating fossil $CO_2$ emissions where the advantage of the high-resolution approach is more evident. The paper is composed as follows: Section 1 provides an introduction; Section 2, the transport model and its adjoint; Section 3 introduces the Prior fluxes, observational dataset, and gridded flux uncertainties; Section 4 gives the formulation of the inverse modelling problem and numerical solution; Section 5 presents simulation results and discussion, which is followed by the Summary and Conclusions.

## 2    The coupled tracer transport model, its adjoint and the implementation

For the simulation of the $CO_2$ transport in the atmosphere, we used the coupled Eulerian-Lagrangian model NIES-TM-FLEXPART, which is a modified version of the model described by Belikov et al. (2016). The coupled transport model is computationally more efficient in comparison to the Eulerian model if both models are run on the same spatial resolution. Belikov et al., (2016) confirmed that the coupled model with a Lagrangian model resolution of 1°×1° performs similarly when coupled with Eulerian model at either 1.25°×1.25° or 2.5°×2.5° resolution, and performance degradation  was only seen when using a 10°×10° resolution Eulerian model. The coupled model consists of NIES-TM v08.1i with a horizontal resolution of 2.5°×2.5° and 32 hybrid-isentropic vertical levels (Belikov et al., 2013) and the FLEXPART model v.8.0 (Stohl et al., 2005) run in backward mode with a surface flux resolution of 0.1°×0.1°. Both models use the Japan 25-year reanalysis (JRA-25)/JMA Climate Data Assimilation System (JCDAS) meteorology (Onogi et al., 2007), with 40 vertical levels interpolated to a 1.25°×1.25° grid. The use of low-resolution wind fields for high-resolution transport is better justified for cases of nearly geostrophic flow over flat terrain, as discussed by Ganshin et al., (2012). It should be useful in the future to adapt this modelling framework to using reanalyses recently made available at 0.25-0.3° resolution, even if the tests with higher resolution winds by Ware et al., (2019) did not show large improvement over lower resolution.

The coupled transport model was derived from the Global Eulerian-Lagrangian Coupled Atmospheric transport model (GELCA) (Ganshin et al., 2012; Zhuravlev et al., 2013; Shirai et al., 2017). To facilitate model application in our iterative

inversion algorithm, all the components of the model – Eulerian model and the coupler are integrated into one executable (online coupling) as described in Belikov et al., (2016), while the original GELCA model implements Eulerian and Lagrangian components sequentially, and then applies the coupler (off-line coupling). The changes in the current version with respect to the version presented by Belikov et al. (2016) include an adjoint code derivation for model components

using the adjoint code compiler Tapenade (Hascoet and Pascual, 2013), instead of using the TAF compiler (Giering and Kaminski, 2003). Additionally, the indexing and sorting algorithms for the transport matrix were revised to allow efficient memory use for processing large matrices of LPDM-driven responses to surface fluxes arising in the case of high-resolution surface fluxes and a large number of observations, especially when using satellite data. A manually derived adjoint of the NIES-TM v08.1i is used as in Belikov et al. (2016), due to its computational efficiency. In the version of the model that

includes manually coded adjoint, only the second-order van Leer algorithm (van Leer, 1977) is implemented, as opposed to the third-order algorithm typically used in forward models (Belikov et al., 2013).

## 3    Prior fluxes, flux uncertainties and observations

Prior $CO_2$ fluxes, were prepared as a combination of monthly-varying fossil fuel emissions by the Open-Data Inventory for Anthropogenic Carbon dioxide (ODIAC), available at a global 30 arc second resolution (e. g.  Oda et al., 2018), ocean-atmosphere exchanges by the Ocean Tracer Transport Model (OTTM) 4D-var assimilation system, available at a 1° resolution (Valsala and Maksyutov, 2010), daily $CO_2$ emissions by biomass burning by the Global Fire Assimilation System (GFAS) dataset provided by the Copernicus services at a 0.1° resolution (Kaiser et al., 2012), and a daily varying

climatology of terrestrial biospheric $CO_2$ exchange simulated by the optimized Vegetation Integrative SImulator for Trace gases (VISIT) model (Ito, 2010; Saito et al., 2014). Figure 1 presents samples of the four prior flux components (fossil, vegetation, biomass burning and ocean) used in the forward simulation.

### 3.1    Emissions from fossil fuel

For the fossil fuel $CO_2$ emissions (emissions due to fossil fuel combustion and cement manufacturing), we used the ODIAC

data product (Oda and Maksyutov, 2011, 2015; Oda et al., 2018) at a 0.1°×0.1° resolution on monthly basis. The version 2016 of the ODIAC data product (ODIAC2016, Oda et al., 2018) is based on global and national emission estimates and monthly estimates made by the Carbon Dioxide Information Analysis Center (CDIAC) (Boden et al., 2016; Andres et al., 2011). For spatial disaggregation it uses the emission data for powerplant emissions by the CARbon Monitoring and Action (CARMA) database (Wheeler and Ummel, 2008), while the rest of the national total emissions on land were distributed

using spatial patterns provided by night-time lights data collected by the Defence Meteorological Satellite Program (DMSP) satellites (Elvidge et al., 1999). The ODIAC fluxes were aggregated to a 0.1° resolution from the high-resolution ODIAC data (1×1 km). The ODIAC emission product is suitable for this kind of study because the global total emission is constrained by updated estimates while providing a high-resolution emission estimate. Thus, it can be applied to carbon budget problems across different scales.

## 3.2    Terrestrial biosphere fluxes

$CO_2$ fluxes by the terrestrial biosphere at a resolution of 0.1° were constructed using a vegetation mosaic approach, combining the vegetation map data by synergetic land cover product (SYNMAP) dataset (Jung et al., 2006), available at a 30 arc second resolution, with terrestrial biospheric $CO_2$ exchanges simulated by optimized VISIT model (Saito et al., 2014) for each vegetation type in every 0.5° grid at a daily time step. The area fraction of each vegetation type is derived from SYNMAP data for each 0.1° grid. The $CO_2$ net ecosystem exchange (NEE) fluxes on a 0.1° grid were prepared by combining the vegetation type-specific fluxes with vegetation area fraction data on a 0.1° grid. By averaging the daily flux data for the period of 2000-2005 the flux climatology was derived for use in the recent years (after 2010) when the VISIT model simulation based on JRA-25-JCDAS reanalysis data is not available. Although the use of climatology in place of original fluxes degrades the prior, the posterior fluxes show significant departures from prior, thus reducing the impact of missing the prior variations. The diurnal cycle was not resolved as it requires producing additionally the gross primary production and ecosystem respiration. To estimate the effect of excluding the diurnal cycle in the prior fluxes, for our selected time of sampling the observations, we compared $CO_2$ concentrations simulated with diurnally varying fluxes at hourly time step with those made with daily mean fluxes produced by the Simple Biosphere model (SiB) model for 2002-2003 (Denning et al, 1996) as used in Transcom continuous intercomparison (Law et al, 2008). The results show that for background monitoring sites the difference is not significant (below 0.1 ppm), similar to the result by Denning et al, (1996). For continental sites, the difference between the two simulations was combined into four seasonal values, and the data for the season with the largest difference were shown in Figure A1. Positive bias by simulation with daily constant flux with respect to diurnally varying fluxes is in the order of 0.5 to 1 ppm, and it is larger during the middle of the growing season. Inclusion of the diurnally varying fluxes in place of daily mean has the potential to change seasonality of posterior fluxes by inversion in a favourable direction, as there are regions where flux seasonality is somewhat stronger than expected (Section 5.2).

### 3.3 Emissions from biomass burning

Daily biomass burning $CO_2$ emissions by the Global Fire Assimilation System (GFAS) dataset rely on assimilating Fire Radiative Power (FRP) observations from the MODIS instruments onboard the Terra and Aqua satellites (Kaiser et al., 2012). The fire emissions at 0.1° resolution are calculated from FRP with land cover-specific conversion factors compiled
from a literature survey. The GFAS system adds corrections for observation gaps in the observations, and filters spurious FRP observations of volcanoes, gas flares, and other sources. The fluxes are input to the model at the surface, which may lead to underestimation of the injection height for strong burning events and occasional overestimation of biomass burning signals simulated at surface stations.

### 3.4 Oceanic exchange flux

The air-sea $CO_2$ flux component for the flux inversion used an optimized estimate of oceanic $CO_2$ fluxes by Valsala and Maksyutov (2010). The dataset was constructed with variational assimilation of the observed partial pressure of surface ocean $CO_2$ ($pCO_2$) available in Takahashi et al. (2017) database into the OTTM (Valsala et al., 2008), coupled with a simple one-component ecosystem model. The assimilation consists of a variational optimization method that minimizes the model-observation differences of the surface ocean dissolved inorganic carbon (DIC) (or $pCO_2$) within the two-month
time window. The OTTM model fluxes produced on a $1° \times 1°$ grid at monthly time step were interpolated to a $0.1° \times 0.1°$ grid, taking into account the land fraction map derived from the 1 km resolution MODIS landcover product.

### 3.5 Flux uncertainties for land and ocean.

$CO_2$ flux uncertainties are needed for both land and ocean regions. Climatological, monthly-varying flux uncertainties for land were set to 20% of the MODIS gross primary productivity (GPP) product (MOD17A2) available on a 0.05° grid at a
monthly resolution (Running et al., 2004). Oceanic flux uncertainties were based on the sum of standard deviation of the OTTM assimilated flux from climatology by Takahashi et al. (2009), and the monthly variance of the interannually-varying OTTM fluxes (Valsala and Maksyutov, 2010), with a minimum value of $0.02 \text{ gCm}^{-2}\text{day}^{-1}$, in the same way as in the lower spatial resolution inverse model by Maksyutov et al. (2013). Oceanic flux uncertainties were first estimated on a $1° \times 1°$ resolution at a monthly time step and then interpolated to a $0.1° \times 0.1°$ grid, with the same procedure as for the oceanic
fluxes.

### 3.6 Atmospheric CO₂ observations.

We used $CO_2$ observation data distributed as the ObsPack-$CO_2$ GLOBALVIEWplus v2.1 (Cooperative Global Atmospheric Data Integration Project, 2016). The data from the flask sites were used as an average concentration for a pair of flasks. Afternoon (15:00 to 16:00 local time) average concentrations were used for continuous observations over land and for remote background observation sites. For the continuous mountain top observations, we used early morning observations (05:00 to 06:00 local time). The geographical local time was used, as defined by UTC with a longitude-dependent offset. The list of the observation locations with ObsPack site ID, site names, data providers, and data references appears in Table A1 in the Appendix, accompanied by a site map in Figure A2. The aircraft observational data collected by NOAA Aircraft Program at Briggsdale, Colorado (CAR), Cape May, New Jersey (CMA), Dahlen, North Dakota (DND), Homer, Illinois (HIL), Worcester, Massachusetts, (NHA), Poker Flats, Alaska (PFA), Rarotonga, (RTA), Charleston, South Carolina (SCA), Sinton, Texas (TGC) (Sweeney et al., 2015), and also by the Comprehensive Observation Network for TRace gases by AIrLiner (CONTRAIL) project over West Pacific (CON) (Machida et al., 2008) were grouped into averages for each 1 km altitude bin, altitude counted from sea level. Within the 1 km altitude range, the average value of both concentration and the altitude was taken. Aircraft observations were not assimilated, only intended for use in the validation of the results.

## 4 Inverse modelling algorithm

### 4.1 Flux optimization problem

Inverse problem of atmospheric transport is formulated by Enting (2002) as finding the surface fluxes that minimize misfit between the transport model simulation $y_f + H \cdot (x_p + x)$ and the vector of observations $y$, where $y_f$ is the forward simulation without the surface fluxes, $x_p$ is the known prior flux, $x$ is the unknown flux correction, and $H$ represents the transport model. The equation $y = y_f + H \cdot (x_p + x)$ has to be solved for the unknown flux correction $x$, and $x$ is solved for at the transport model grid scale (Kaminski et al., 2001). By introducing the residual misfit vector $r = y - (y_f + H \cdot x_p)$, the problem can be formulated as minimizing a norm of difference $(r - H \cdot x)$ weighted by the data uncertainties. As the observation data alone are not sufficient to uniquely define the solution $x$, an additional regularization is required. By introducing additional constraints on the amplitude and smoothness of the solution, the inverse modelling problem is formulated (Tarantola, 2005) as solving for the optimal value of the vector $x$ at the minimum of a cost function $J(x)$:

$$J(x) = \frac{1}{2}(H \cdot x - r)^T \cdot R^{-1} \cdot (H \cdot x - r) + \frac{1}{2}x^T \cdot B^{-1} \cdot x \qquad (1)$$

where $x$ is the optimized flux, $R$ is the covariance matrix for observations and $B$ is the covariance matrix for surface fluxes. By introducing a decomposition of $B$ as $B = L \cdot L^T$ (construction of matrix $L$ explained in detail in Section 4.2) and a variable substitution $x = L \cdot z$ the second term in Eq. (1) is simplified. At the same time, by assuming that $R$ can be decomposed into $R = \sigma^T \cdot \sigma$, where $\sigma$ is a vector of data uncertainties, and introducing expressions $b = \sigma^{-1} \cdot (r - H \cdot x)$, and $A = \sigma^{-1} \cdot H \cdot L$, the new form of Eq. (1) is introduced:

$$J(z) = \frac{1}{2}((A \cdot z - b)^T(A \cdot z - b) + z^T \cdot z) \qquad (2)$$

The solution minimizing $J(z)$ can be obtained by forcing the derivative $\partial J(z)/\partial z = A^T(A \cdot z - b) + z$ to be zero, which results in

$$(A^T A + I) \cdot z = A^T b \qquad (3)$$

An optimal solution $z$ at the minimum of the cost function $J(z)$ is found iteratively with the Broyden–Fletcher–Goldfarb–Shanno (BFGS) algorithm (Broyden, 1969; Nocedal, 1980), as implemented by Gilbert and Lemarechal (1989). The method requires the ability to accurately estimate the cost function $J(z)$ and its gradient $A^T(A \cdot z - b) + z$, and has modest memory storage demands. Given the solution $z$, the flux correction vector $x$ is then found by reversing the variable substitution as $x = L \cdot z$.

The convergence of the solution may be affected by the accuracy of the adjoint. The result of the duality test defined as the norm of the difference between NIES-TM-FLEXPART forward and adjoint modes estimated as $(< y, H \cdot x > - < H^T \cdot y, x >)/(< y, H \cdot x >)$ was found to be in the order of $10^{-9}$, while for the Lagrangian component based on the receptor sensitivity matrices prepared with FLEXPART, it is about $10^{-15}$ when calculated in double precision (same as in Belikov et al., 2016). The formulation of the minimization problem as presented by Eq. (2) is convenient for the derivation of the flux uncertainties, as it is possible to solve Eq. (3) via the truncated singular value decomposition (SVD) and estimate the regional flux uncertainties based on the derived singular vectors (Meirink et al., 2008). Alternatively, as mentioned by Fisher and Courtier (1995), it is also possible to use the flux increments derived at each iteration of the BFGS algorithm in place of the singular vectors. Although we did not use SVD for constructing the posterior covariances in this study, we tested solving the optimization problem with SVD. We derived SVD of $A^T A$ using a computer code by Wu and Simon (2000), which implements an algorithm proposed by Lanczos (1950) and confirmed that our approach yields practically the same solution as the one obtained with the BFGS algorithm. Lanczos (1950) algorithm is a commonly used SVD technique applied in case of large sparse matrix or a linear operator, when it is impractical to directly make SVD of $A$. A

truncated SVD of $A$ is given by the expression $A \approx U\Sigma V^T$, where $\Sigma$ is the diagonal matrix of $n$ singular values, while $U$ and $V$ are the matrices of left and right singular vectors. Variable substitutions

$$z = V^T s, d = U^T b, \tag{4}$$

transform $z$ into a space of singular vectors $s$ and reduces Eq. (3) to $(\Sigma^T\Sigma + I) \cdot s = \Sigma^T d$ , resulting in a solution

5    $s = \Sigma^T d / (\Sigma^T\Sigma + I)$, (5)

which is evaluated directly, as $\Sigma$ is diagonal. In case of having only $n$ largest singular values, the elements of the solution $s$ are given by $s_i = \lambda_i d_i / (\lambda_i^2 + 1)$, for all $i \leq n$. Once the solution (5) is found, it is taken back to the space of the dimensional fluxes $z$ by applying variable substitutions (4). For fluxes, we have $x = L \cdot z, z = V^T s, d = U^T b$, thus the solution is provided by

10    $x = LV \cdot \frac{\Sigma^T}{(\Sigma^T\Sigma + I)} \cdot U^T b.$ (6)

Another variant of the SVD approach may be more memory efficient in the case of a very large dimension of a flux vector, then applying SVD to $AA^T$ instead of $A^T A$ can save some memory as in a representer method (Bennett, 1992). It gives the same solution as SVD of $A^T A$ using less intermediate memory storage when the dimension of the observation vector $y$ is lower compared to that of the flux vector $x$.

The forward and adjoint mode simulations with the transport model needed to implement the iterative optimization are composed of several steps:

1. Running the Lagrangian model FLEXPART to produce the source-receptor sensitivity matrices. For each observation event, a backward transport simulation with the FLEXPART model is implemented, to produce the

20        surface flux footprints at a $0.1° \times 0.1°$ latitude-longitude resolution and the 3-D concentration field footprint, taken at the end of the backward simulation run (ending at the coupling time of 00:00 GMT). The coupling time is set to be within 2 to 3 days before the observation event. The surface flux sensitivity data are recorded in the unit of ppm (gCm$^{-2}$day$^{-1}$)$^{-1}$. The flux footprints are saved at a daily or hourly timestep, depending on available surface fluxes.

25    2. Running the coupled transport model forward, which includes:

    a. Running the 3-D Eulerian model NIES-TM from the 3-D initial concentration field, with the prescribed surface fluxes. Sampling the 3-D field at model coupling times for each observation according to 3-D concentration field footprints, calculated at the first step by FLEXPART. NIES-TM reads the same $0.1°$ fluxes as the Lagrangian transport model and remaps them onto its $2.5° \times 2.5°$ grid, before including them

30        in the simulation. For each observation event, the fluxes used in Eulerian and Lagrangian components

are separated by the coupling time, so that there is no double-counting of fluxes for the same date in the coupled model simulation.

    b.    Use the two-dimensional (2-D) surface flux footprints prepared with the Lagrangian model to calculate the surface flux contribution to the simulated concentrations for the last 3 days.

    c.    Combining the concentration contributions produced by Eulerian (a) and Lagrangian (b) components to give the total simulated concentration.

3. In the inverse modelling, the transport model is run in three modes:

    a.    The forward model is first run with prescribed prior fluxes, starting from the 3-D initial $CO_2$ concentration field, to calculate the residual misfit (difference between the observation and the model simulation).

    b.    At the inverse modelling/optimization step, only the flux corrections are propagated in the forward model runs, which are optimized to minimize the observation-model misfit. The prescribed prior fluxes are not used (switched off) at this step. The model starts from a zero 3-D initial concentration field and runs forward with flux corrections updated by the optimization algorithm at every iteration, to produce simulated concentrations. Corrections to the 3-D initial concentration field are not estimated, and thus not included in the control vector. Instead, the model is given a spin-up period of three months before the target flux estimation period to adjust the simulated concentration to the observations.

    c.    In the adjoint mode, the adjoint mode atmospheric transport is simulated backward in time starting from the vector of residuals to produce a gradient of the cost function (defined as eq. (1)) with respect to the surface fluxes. Given the gradient, the optimization algorithm provides the new flux corrections field. For convenience, the transport model and its adjoint are implemented as callable procedures suitable for direct communication mode.

Steps 1 is carried out the same way as in other versions of the coupled transport model (Zhuravlev et al., 2013; Shirai et al., 2017). In steps 2 and 3, the procedure of running the forward and adjoint model is organised differently. At the beginning of the transport model runs, all the data prepared by the Lagrangian model are stored in the computer memory, in order to save the time required for reading and re-sorting the data at each iteration. The fraction of the CPU time spent on running the Eulerian component of the coupled transport model is 82%, on the Lagrangian component 1%, and on covariance 17%.

To create the initial concentration field, we used a 3-D snapshot of $CO_2$ concentration for the same day from a simulation of the previous year, which is already optimized (usually Oct 1[st], or Jan 1[st]). When such simulation is not available, we

took snapshot from an available year and correct it globally for the concentration difference between these years using the NOAA monthly mean data for the South Pole as representative for the global mean concentration. When the optimized fields are not available, the output of the multiyear spin-up simulation is used, with the same adjustment to the South Pole observations.

## 4.2 Implementation of covariance matrices $L$ and $B$.

We optimized surface flux fields separately for two sets of fluxes in every grid globally, for land and ocean regions, following the approaches by Meirink et al. (2008) and Basu et al. (2013), who suggested optimizing for global surface flux fields separately for each optimized flux category. Separating the total flux into independent flux categories, each with its own flux uncertainty pattern, results in using homogenous spatial covariance matrices, significantly simplifying the coding of the matrix $B$. The matrix $B$ can be given as the product of a diagonal matrix of flux uncertainties and a matrix with 1.0 as diagonal elements, while non-diagonal elements are exponentially declining with the squared distance between grid points (Meirink et al., 2008). In practice, an extra scaling of the uncertainty is needed for balancing the constraint on fluxes with the data uncertainty, which also impacts the regional flux uncertainties. Several empirical methods are in use, where the tuning parameters are a horizontal scale (Meirink et al., 2008) and an uncertainty multiplier (Chevallier et al., 2005; Rodenbeck, 2005). In our $B$ matrix design, we follow Meirink et al. (2008) in representing $B$ matrix as the multiple of the non-dimensional covariance matrix $C$ and the diagonal matrix of the flux uncertainty $D$ as $B = D^T \cdot C \cdot D$. $C$ matrix is commonly implemented as a band matrix with non-diagonal elements declining as $\sim \exp(-x^2/l^2)$ with the distance $x$ between the grid cells, as in 2-D spline algorithms (Wahba and Wendelberger, 1980). Multiplication by the matrix $C$ becomes computationally costly at a high spatial resolution in cases where the correlation distance $l$ is much larger than the size of the model grid. The correlation distance used here is 500 km for land and ocean, and two weeks in time. The rationale of applying a correlation distance of 500 km in the case of a regional inversion over the continental USA with a model grid size of 40 km was discussed by Schuh et al. (2010). In that case, the use of an implicit diffusion with a directional splitting to approximate the Gaussian shape appears to be computationally more efficient than the direct application of the Gaussian-shaped smoothing function, as the number of floating-point operations per grid point does not grow with the ratio of the correlation distance $l$ to the grid size. The covariance matrix based on the diffusion operator is popular in many ocean data assimilation systems, as a convenient way to deal with coastlines (e. g. Derber and Rosati, 1989; Weaver and Courtier, 2001).

The idea of using the solution of the diffusion equation instead of multiplying a vector by the covariance matrix can be presented briefly in a 1-D case. Consider a discrete problem of multiplying a vector representing a function $g(\lambda)$ on a grid with spacing $\Delta\lambda$ by a symmetric matrix which has diagonal elements equal to one, and non-diagonal ones declining as $\exp\left(-\frac{1}{2}(i\Delta\lambda)^2/d^2\right)$ with distance of $i$ points from the diagonal, where $d$ is covariance length. Its continuous analogue is

an application of a Gaussian-shaped smoother in the form $G(\lambda, \lambda') = \exp\left(-\frac{1}{2}(\lambda - \lambda')^2/d^2\right)$ to a function $g(\lambda)$ as:

$$\tilde{g}(\lambda) = \int_{-l}^{l} \exp\left(-\frac{1}{2}(\lambda - \lambda')^2/d^2\right) g(\lambda')\, d\lambda', \tag{7}$$

where the smoothing window size $l$ should be several times larger than $d$. The expression in Eq. (7) looks exactly like the solution of a one-dimensional diffusion equation

$$\frac{\partial g}{\partial t} - D \frac{\partial^2 g}{\partial \lambda^2} = 0, \tag{8}$$

where $D$ is the diffusivity. The solution of Eq. (8) is given by $\acute{g}(\lambda) = \frac{1}{\sqrt{2\pi p^2}} \int_{-l}^{l} \exp\left(-\frac{1}{2}(\lambda - \lambda')^2/p^2\right) g(\lambda')\, d\lambda'$, where

$p^2 = 2D\Delta t$, $g(\lambda)$ is the initial distribution, and $\Delta t$ is the time step (Crank, 1975). Based on this equivalence, instead of multiplying a vector by the covariance matrix, we solve the discrete form of Eq. (8) by backward-in-time, central-in-space implicit method.

Applying the diffusion operator for the covariance matrix helps to achieve the spatial homogeneity between polar and

equatorial regions, as diffusion produces a theoretically uniform effect on flux fields regardless of the polar singularity. The diffusion operator works as a low-pass filter, selectively suppressing all the wavelengths shorter than the covariance length scale. As we need to construct the covariance matrix $B$ in the form $B = L \cdot L^T$, we choose to construct $L$ first and then derive its transpose $L^T$. The factorization of $L$ is given by $L = u_F \cdot (L_{xy} \otimes L_t) \cdot m$, where $L_t$ is the one-dimensional covariance matrix for time dimension, $\otimes$ is a Kronecker product. We approximate the two-dimensional covariance $L_{xy}$ by

splitting it into two dimensions, latitude and longitude, as in Chua and Bennett, (2001), and apply several iterations of this process. The horizontal covariance $L_{xy}$ is implemented in $N = 3$ iterations of one-dimensional diffusion so that $L_{xy} = (L_x \otimes L_y)^N$, where $L_x$ and $L_y$ are covariance operators for longitude and latitude directions respectively, while $u_F$ is the diagonal matrix of flux uncertainty for each grid cell and each flux category (land and ocean), and $m$ is the diagonal matrix of a map factor which is introduced to scale the contributions to the cost function by model grid area, with diagonal elements

given by $m = \cos^{-1/2}\theta$ (where $\theta$ is latitude).

This design of covariance operator helps to preserve the high-resolution structure of the resultant flux corrections, given by $x = L \cdot z = u_F \cdot (L_{xy} \otimes L_t) \cdot m \cdot z$, as it can be factored into a multiple of uncertainty $u_F$ and scaling factor $S = (L_{xy} \otimes L_t) \cdot m \cdot z$ as $x = u_F \cdot S$. While the scaling factor $S$ is smoothed with a covariance length of 500 km, the

original structure of the spatial heterogeneity of surface flux uncertainty $u_F$ is still preserved at the original high-resolution in the optimized flux corrections $x$.

The adjoint operators $L_x^T$ and $L_y^T$ are derived by applying the adjoint code compiler Tapenade (Hascoet and Pascual, 2013) to the Fortran code of modules that approximate the operators $L_x$ and $L_y$ by implicit diffusion. $L_t$ and its transpose $L_t^T$ are of lower dimension and are designed as in Meirink et al. (2008) by deriving the square root of the Gaussian-shaped time covariance matrix with direct SVD (Press et al., 1992).

A notable merit of the algorithm is that it minimizes the use of the computer memory, avoiding allocation of the memory space larger than several times the dimension of the observation and flux vectors, making it suitable for ingesting large amounts of surface and space-based observations. It should be mentioned that the computer memory demand for accommodating the surface flux sensitivity matrices for massive space-based observations can be a limiting factor as discussed by Miller et al. (2020).

## 4.3    Inversion setup

The combination of the coupled transport model NIES-TM-FLEXPART (as described in Section 2) with the variational optimization algorithm (Sections 4.1-4.2) constitutes the inverse modelling system NIES-TM-FLEXPART-VAR (NTFVAR). We tested the inversion algorithm presented in previous sections with the problem of finding the best fit to the $CO_2$ observations provide by ObsPack dataset by optimizing the corrections to the land and ocean fluxes. By the design of our inverse modelling system, we produced smoothed fields of scaling factors that are multiplied by the fine-resolution flux uncertainty fields to give flux corrections. We derived the surface $CO_2$ flux corrections at a 0.1° resolution and biweekly time step. Our purpose is to demonstrate that we can optimize fluxes to improve the fit to the observations using iterative optimization procedure, based on high-resolution coupled transport model and its adjoint. Our report is limited to the technical development towards achieving the capability of estimating anthropogenic $CO_2$ emissions based on atmospheric observations, and we do not elaborate on the impact of simulating the tracer transport at a high resolution on the quality of the optimized natural fluxes, which requires an additional study. The flux optimization was applied to a short time-window of 18 months, for each optimized year, and the simulation starts on October 1[st], three months ahead of the target year. A spin-up period of three months is given to let the inversion adjust the modeled concentration to the observations so that a balance is achieved between fluxes, concentrations and concentration trends. The simulation is continued until reaching the limit of 45 cost function gradient calls, by that time M1QN3 procedure by Gilbert and Lamarechal (1989) is able to complete 30 iterations. Figure A3 in the Appendix presents the cost function reduction in the

case of optimizing fluxes for 2011 and completing 61 gradient calls. The cost function reduction declines nearly exponentially, by almost 3 times for each 10 gradient calls completed. The relative improvement between 41 and 61 gradient calls is 1.5% of the total reduction from the first to the 61 gradient calls. We optimized fluxes for three years from 2010 to 2012 and analyzed the simulated concentration fit at the observation sites. The average root mean squared misfits (RMSE) between the optimized concentrations and the observations are compared with a forward simulation with prior fluxes and optimized simulation. For evaluation, we used statistics of the optimized simulations by the operational NOAA's CarbonTracker inverse modelling system (ObsPack_co2_1_CARBONTRACKER_CT2017_2018-05-02; Peters et al., 2007).

## 5    Results and discussion.

### 5.1 Analysis of the posterior model fit to the observations

We compared the results of the forward simulation with prior and optimized fluxes with the processed observations for ground observation sites, as shown in Table A1, and airborne vertical profiles were used for an independent validation (Table A2). Figure 2 shows the observations with forward (prior) and optimized simulations at Barrow (BRW), Jungfraujoch (JFJ), Wisconsin (LEF), Pallas (PAL), Yonagunijima (YON), and Syowa (SYO). The optimization yielded improved seasonal variations of the simulated concentration, including the phase and the amplitude at most sites. At SYO we found synoptic scale variations with an amplitude in the order of few tenths of a ppm, that were to a large extent, captured by the model. Plots for BRW and JFJ show the ability of the inversion to correct the seasonal cycle, while the difference between model and observations in the southern hemisphere (SYO) is contributed by interannual variations of the carbon cycle. The model-observation mismatch (RMSE) for surface sites included in the ObsPack is presented in Figure 3 for forward and optimized simulation and mean bias for optimized data. The model was able to reduce the model to observation mismatch for most background sites, where the seasonal cycle is affected mostly by natural terrestrial and oceanic fluxes, while the average reduction of the mismatch from forward to optimized simulation is 14%, defined as the mean ratio of optimized mismatch to forward mismatch taken for each site. The reason for the relatively small reduction is the addition of climatological flux corrections to the prior, estimated by inverse modelling of two years of data, 2009 and 2010. As a result, the inversion starts from the initial flux distributions already adjusted to fit the seasonal cycle of the observed concentration. The correction for the difference in global concentration trend between years is not made, thus there are visible differences between prior and optimized simulations in the southern hemispheric background sites. At most of the Antarctic sites, the mean posterior (after optimization) mismatch (reported as RMSE) is in the order of 0.2 ppm. Over the land, closer to anthropogenic sources, there is a less relative reduction of mismatch on an annual mean scale.

One of the reasons for seeing little improvement is keeping fossil $CO_2$ emissions fixed and optimizing only the natural fluxes (while the strong signal from fossil emission is not affected by flux corrections). Another possible contributor to the large mismatch over land is neglecting the diurnal cycle under the assumption of using only observations at well-mixed conditions, and also the limited ability of the low-resolution reanalysis dataset to capture frontal processes in the extratropical continental atmosphere, as discussed by Parazzoo et al. (2011). The mean mismatch was reduced from 2.60 ppm to 2.42 ppm by the flux optimization, while the mean mismatch to uncertainty ratio decreases after optimization by 19% from 0.94 to 0.78. The mean correlation between modelled and observed data improves from $r^2$ =0.43 ($r^2$ - coefficient of determination) for the simulation driven by prior fluxes to $r^2$ =0.59 for the optimized simulation. To remove the effect of the interannual $CO_2$ growth on $CO_2$ variabilities, the mean growth trend was subtracted from data before estimating the $r^2$.

Figure 3 also shows, for a comparison purpose, the statistics of the average misfit for the optimized simulation by CarbonTracker, for the same period and same monitoring stations. The comparison is useful for understanding the strength and weaknesses of the inversion system presented here. Over the background monitoring sites, the high-resolution model does not show any advantage over CarbonTracker in terms of the fit between optimized model simulation and observations, which may indicate better performance by the Eulerian model TM5 used in CarbonTracker. On the other hand, several sites where the high-resolution model shows better fits to observations over CarbonTracker are located inland or near the coast, closer to anthropogenic and biogenic sources. A smaller misfit was achieved by the high-resolution model at Key Biscaine (KEY), Baring Head (BHD), Marianna island (GMI), and Cape Kumukahi (KUM), among others, which can be attributed to coastal/island locations, while there is little or no advantage at mountain sites like Mauna Loa (MLO) or Jungfraujoch (JFJ). This result may be influenced by differences in the model physics between NIES-TM-FLEXPART and TM5 in the lower troposphere, near the top of the boundary layer, and in shallow cumuli. The mismatch (RMSE) between our optimized model and observations for the 102 sites used in the inversion is only 4% lower on average than that by CarbonTracker. It is not yet clear if there is a systematic advantage of one or the other system in any particular site category, other than for coastal/island sites mentioned above. For the average misfit comparison, all data, both assimilated and not assimilated, are included for sites shown in Figure 3. The results for CGO were not counted, due to the use of different datasets, as our system used only the NOAA flask data, which underwent background selection (by wind direction) at the time of sampling.

As an independent validation, a comparison of the unoptimized and optimized simulation to the vertical profile data is shown in Figure 4. For each vertical profile site, the observations were grouped by altitude, at a 1 km interval. The altitude code (e. g. 005, 015, 025, 035, …) to be added to the site identifier was constructed as the altitude of midlevel multiplied by 10. The observations at PFA (Poker Fat Alaska) between the surface and 1 km were grouped as PFA005 (mid altitude

0.5 km), while those in 5 to 6 km range were designated as PFA055 (mid-altitude 5.5 km). As for optimized surface data in Figure 3, we show RMSE for forward simulation with prior fluxes, optimized simulation, and CarbonTracker, and mean bias for optimized data. CarbonTracker shows a better fit at most of the altitude ranges except for the lowest 1 km where the results shown by the two systems were similar. Concurrently, the mean correlation between modelled and observed
data did not improve from prior ($r^2$ =0.70) to optimized simulation ($r^2$ =0.63), while mean RMSE declined a little from 1.86 ppm to 1.85 ppm. The comparison to CarbonTracker (CT2017), with a mean RMSE of 1.53 ppm, suggests that the free tropospheric performance of our system can be improved by implementing more detailed vertical mixing processes in the Lagrangian and Eulerian component models.

## 5.2 Comparison of prior and posterior fluxes

As mentioned in section 4.2, the flux corrections estimated by the inverse model showed fine scale features despite using large covariance length, because those were made of the high-resolution data uncertainty multiplied by the smooth fields of scaling factor, estimated separately for each of the optimized flux categories - land biosphere and ocean. Examples of the flux corrections and posterior fluxes (excluding fossil emissions) are presented in Figure 5. The flux corrections and fluxes are shown in Figure 5 for one month (Aug 2011) as an illustration, and they are not representative of a seasonal or
climatological mean. The sign of the flux corrections changed from positive (source) in the eastern side (continental China) to negative (sink) over the Russian coast and Japanese islands, while the posterior fluxes appeared as a terrestrial sink all over the area. The flux adjustment was driven by the fit to nearby observations made over Korea and Japan.
To illustrate the change of fluxes from prior to posterior estimate by the inversion at the scale of large aggregated regions, the monthly mean fluxes (excluding fossil emissions) averaged for 3 years 2010-2012 are plotted in Figure 6 for eight
selected Transcom regions (as defined by Gurney et al. 2002, see map in Figure A2). The plots include prior, optimized, and, for reference, optimized fluxes by CarbonTracker (CT2017). For some regions, the posterior is close to the prior, which is often the case when there are too few observations in the region to drive the corrections to prior fluxes. Boreal North America (region 1), Temperate North America (2), and Europe (11) are better constrained by observations, while North Africa (5), South Africa (6), Temperate Asia (8), South-East Asia (9) and Boreal Asia (7) are less constrained. The
optimized flux is similar to the prior for Africa (5, 6), South-East Asia (10), and Temperate Asia (8), while there is a substantial adjustment for Boreal Asia (7), which seems to be adjusted to fit the observations outside the region. For both boreal regions, the prior flux seasonality appears weaker than in both posterior and CarbonTracker, which could indicate a problem with vegetation type mapping in higher resolution version of the prior flux model. For regions 1, 6, 7, and 11, the corrected fluxes are closer to CarbonTracker, and for Temperate North America, Temperate Asia, and North Africa the
amplitude of flux seasonality is estimated to be stronger, which can be caused by stronger vertical/horizontal mixing in

transport model as compared to the transport in CarbonTracker. A more detailed comparison with other inverse model results and independent estimates (e.g. by Jung et al., 2020) should be made after improving the inversion setup, notably, improving the transport model meteorology, seasonality, and diurnal cycle in prior fluxes and seasonality in prior flux uncertainties.

**6 Summary and conclusions**

A grid-based $CO_2$ flux inversion system that is suitable for inverse estimation of the surface fluxes at a biweekly time step and a 0.1° spatial resolution, was developed. To implement the high-resolution simulation capability, several developments were completed. High-resolution prior fluxes were prepared for three surface flux categories: fossil emissions by the ODIAC dataset are based on the point source database and nightlights, biomass burning emissions (GFAS) are based on MODIS observations of fire radiative power and biosphere exchange is based on the mosaic representation of landcover and process-based VISIT model simulation. A high-resolution atmospheric transport for a global set of observations was achieved by combining short-term simulations with the high-resolution Lagrangian model FLEXPART with a global three-dimensional simulation with the medium-resolution Eulerian model NIES-TM. The use of variational optimization with a gradient-based method in the inversion helps to avoid the need for inverting large matrices with dimensions dictated by the number of optimized grid fluxes or the number of observations. Accordingly, the adjoint of the coupled transport model was developed to apply the variational optimization. A computationally efficient implementation of flux error covariance operator is achieved by using an implicit diffusion algorithm. Overall, the presented algorithm demonstrated the feasibility of the high-resolution inverse modelling at the global scale, extending the capabilities achieved by regional high-resolution modelling approaches used for estimating the national greenhouse gas emissions for comparison with the national greenhouse gas inventories. A comparison of the optimized simulation to the observations showed some improvements over lower resolution CarbonTracker model for some continental and coastal observation sites, located closer to anthropogenic emissions and strong biospheric fluxes, but also demonstrates the need for further improvement of the inverse modelling system components. Transport model errors can be reduced by improving transport modelling algorithms in the Eulerian and Lagrangian models and using a combination of recent higher resolution reanalysis data with high-resolution wind data simulations by regional models in the regions of interest. Our inverse modelling algorithm can be further improved by tuning the uncertainty scaling, and spatial and temporal covariance distances. Prior fluxes can be improved by developing high-resolution diurnally varying biospheric fluxes, developing a more detailed fossil emission inventory, and updating estimates of biomass burning and oceanic fluxes.

**Code and data availability.**

The inverse model and forward transport model code can be made available to potential research collaborators upon reasonable request. The ObsPack dataset is available from NOAA/ESRL (https://www.esrl.noaa.gov/gmd/ccgg/obspack/). The ODIAC fossil fuel emission dataset is available from the Global Environmental Database hosted by CGER/NIES
(http://db.cger.nies.go.jp/dataset/ODIAC/).

**Author contributions.**

SM developed the inverse and transport model algorithms and model codes, ran the model, analysed the results, and wrote the manuscript. TO developed the anthropogenic emission inventory, MS developed biospheric flux dataset, JK provided the biomass burning emission fluxes, VV prepared the oceanic $CO_2$ fluxes. RJ contributed to model testing and data
preparation, DB contributed to the development of the NIES transport model, coupled model and coupled adjoint, RZ and AG developed the Lagrangian response simulation system based on FLEXPART model. AA, LC, ED, LH, TM, TN, MR, RL, CS, DR contributed the observational data. All the authors contributed to the development of the manuscript.

**Competing interests.**

The authors declare no competing interests.

**Acknowledgements**

The authors acknowledge the use of computing resources at the National Institute for Environmental Studies (NIES) super-computer facility and support from the GOSAT project, project leaders Tatsuya Yokota and Tsuneo Matsunaga, the Ministry of the Environment (MOE) Japan, MRV grant to NIES, grants by the Ministry of Education, Culture, Sports, Science and Technology (MEXT) of Japan to GRENE and ArCS projects. The model development benefitted from fruitful
discussions with Frederic Chevallier, David Baker, Peter Rayner, Aki Tsuruta, Fenjuan Wang, Prabir Patra, John Miller, Misa Ishizawa, Tomoko Shirai and the members of the TRANSCOM project. Lorna Nayagam provided her technical assistance to our model testing and dataset developments. Authors appreciate contribution of NOAA CarbonTracker data provided by Andy Jacobson and colleagues. The ObsPack was compiled and distributed by NOAA/ESRL. The authors are grateful to the ObsPack data contributors at NOAA GMD, the Environment and Climate Change Canada and other
institutions worldwide, including Tuula Aalto, Shuji Aoki, Gordon Brailsford, Marc L. Fischer, Grant Forster, Angel J.

Gomez-Pelaez, Juha Hatakka, Arjan Hensen, Casper Labuschagne, Ralph Keeling, Paul Krummel, Markus Leuenberger, Andrew Manning, Kathryn McKain, Frank Meinhardt, Harro Meijer, Shinji Morimoto, Jaroslaw Necki, Paul Steele, Britton Stephens, Atsushi Takizawa, Pieter Tans, Kirk Thoning and their colleagues.

**Appendix**

5  Table A1. List of the observation sites included in the ObsPack dataset

| Site ID | Lat | Lon. | Site name | Lab name | Sampling | Reference |
|---------|-----|------|-----------|----------|----------|-----------|
| ALT | 82.45 | -62.51 | Alert | EC | insitu | Worthy et al. 2003 |
| ALT | 82.45 | -62.51 | Alert | NOAA | flask | Conway et al. 1994 |
| AMS | -37.8 | 77.54 | Amsterdam Island | LSCE | insitu | Gaudry et al. 1991 |
| AMT | 45.03 | -68.68 | Argyle | NOAA | insitu | Andrews et al. 2014 |
| ARA | -23.86 | 148.47 | Arcturus | CSIRO | flask | Francey et al. 2003 |
| ASC | -7.97 | -14.40 | Ascension Island | NOAA | flask | Conway et al. 1994 |
| ASK | 23.26 | 5.63 | Assekrem | NOAA | flask | Conway et al. 1994 |
| AZR | 38.77 | -27.38 | Terceira Island | NOAA | flask | Conway et al. 1994 |
| BAO | 40.05 | -105.00 | Boulder Atmospheric Observatory | NOAA | insitu | Andrews et al. 2014 |
| BCK | -116.1 | 62.80 | Bechoko | EC | insitu | Worthy et al. 2003 |
| BHD | -41.41 | 174.87 | Baring Head Station | NOAA | flask | Conway et al. 1994 |
| BHD | -41.41 | 174.87 | Baring Head Station | NIWA | insitu | Brailsford et al. 2012 |
| BMW | 32.27 | -64.88 | Tudor Hill | NOAA | flask | Conway et al. 1994 |
| BRA | 51.2 | -104.7 | Bratt's Lake Saskatchewan | EC | insitu | Worthy et al. 2003 |
| BRW | 71.32 | -156.61 | Barrow | NOAA | flask | Conway et al. 1994 |
| BRW | 71.32 | -156.61 | Barrow | NOAA | insitu | Peterson et al. 1986 |
| CBA | 55.21 | -162.72 | Cold Bay | NOAA | flask | Conway et al. 1994 |
| CES | 51.97 | 4.93 | Cesar | ECN | insitu | Vermeulen et al. 2011 |
| CGO | -40.68 | 144.69 | Cape Grim | NOAA | flask | Conway et al. 1994 |
| CHL | 58.75 | -94.07 | Churchill | EC | insitu | Worthy et al. 2003 |
| CHR | 1.70 | -157.15 | Christmas Island | NOAA | flask | Conway et al. 1994 |

| CIB | 41.81 | -4.93 | Centro de Investigacion de la Baja Atmosfera | NOAA | flask | Conway et al. 1994 |
|-----|-------|-------|---------------------------------------------|------|-------|--------------------|
| CPT | -34.35 | 18.49 | Cape Point | NOAA | flask | Conway et al. 1994 |
| CPT | -34.35 | 18.49 | Cape Point | SAWS | insitu | Brunke et al. 2004 |
| CRI | 15.08 | 73.83 | Cape Rama | CSIRO | flask | Francey et al. 2003 |
| CRZ | -46.43 | 51.85 | Crozet Island | NOAA | flask | Conway et al. 1994 |
| CYA | -66.28 | 110.52 | Casey | CSIRO | flask | Francey et al. 2003 |
| DRP | -59.12 | -63.63 | Drake Passage | NOAA | ship flask | Conway et al. 1994 |
| EGB | 44.23 | -79.78 | Egbert | EC | insitu | Worthy et al. 2003 |
| EIC | -27.15 | -109.45 | Easter Island | NOAA | flask | Conway et al. 1994 |
| ESP | 49.38 | -126.54 | Estevan Point | EC | insitu | Worthy et al. 2003 |
| EST | 51.66 | -110.21 | Esther | EC | insitu | Worthy et al. 2003 |
| ETL | 54.35 | -104.98 | East Trout Lake | EC | insitu | Worthy et al. 2003 |
| FSD | 49.88 | -81.57 | Fraserdale | EC | insitu | Worthy et al. 2003 |
| GMI | 13.39 | 144.66 | Mariana Islands | NOAA | flask | Conway et al. 1994 |
| GPA | -12.25 | 131.04 | Gunn Point | CSIRO | flask | Francey et al. 2003 |
| HBA | -75.61 | -26.21 | Halley Station | NOAA | flask | Conway et al. 1994 |
| HDP | 40.56 | -111.65 | Hidden Peak (Snowbird) | NCAR | insitu | Stephens et al. 2011 |
| HPB | 47.80 | 11.02 | Hohenpeissenberg | NOAA | flask | Conway et al. 1994 |
| HUN | 46.95 | 16.65 | Hegyhatsal | HMS | insitu | Haszpra et al. 2001 |
| HUN | 46.95 | 16.65 | Hegyhatsal | NOAA | flask | Conway et al. 1994 |
| INX | -86.02 | 39.79 | INFLUX (Indianapolis Flux Experiment) | NOAA | flask | Conway et al. 1994 |
| IZO | 28.31 | -16.50 | Izana | NOAA | flask | Conway et al. 1994 |
| IZO | 28.31 | -16.50 | Izana | AEMET | insitu | Gomez-Pelaez et al. 2011 |
| JFJ | 46.55 | 7.99 | Jungfraujoch | KUP | insitu | Uglietti et al. 2011 |
| KAS | 49.23 | 19.98 | Kasprowy Wierch | AGH | insitu | Necki et al. 2003 |
| KEY | 25.66 | -80.16 | Key Biscayne | NOAA | flask | Conway et al. 1994 |
| KUM | 19.52 | -154.82 | Cape Kumukahi | NOAA | flask | Conway et al. 1994 |

| LEF | 45.95 | -90.27 | Park Falls | NOAA | insitu | Andrews et al. 2014 |
|-----|-------|--------|------------|------|--------|---------------------|
| LJO | 32.87 | -117.26 | La Jolla | SIO | flask | Keeling et al. 2005 |
| LLB | 54.95 | -112.45 | Lac La Biche | EC | insitu | Worthy et al. 2003 |
| LLB | 54.95 | -112.45 | Lac La Biche | NOAA | flask | Conway et al. 1994 |
| LMP | 35.52 | 12.62 | Lampedusa | NOAA | flask | Conway et al. 1994 |
| LUT | 53.4 | 6.35 | Lutjewad | RUG | insitu | van der Laan et al. 2009 |
| MAA | -67.62 | 62.87 | Mawson Station | CSIRO | flask | Francey et al. 2003 |
| MEX | 18.98 | -97.31 | High Altitude Global Climate Observation Center | NOAA | flask | Conway et al. 1994 |
| MHD | 53.33 | -9.9 | Mace Head | NOAA | flask | Conway et al. 1994 |
| MHD | 53.33 | -9.9 | Mace Head | LSCE | insitu | Ramonet et al. 2010 |
| MID | 28.21 | -177.38 | Sand Island | NOAA | flask | Conway et al. 1994 |
| MLO | 19.54 | -155.58 | Mauna Loa | NOAA | flask | Conway et al. 1994 |
| MLO | 19.54 | -155.58 | Mauna Loa | NOAA | insitu | Thoning et al. 1989 |
| MNM | 24.28 | 153.98 | Minamitorishima | JMA | insitu | Tsutsumi et al. 2005 |
| MQA | -54.48 | 158.97 | Macquarie Island | CSIRO | flask | Francey et al. 2003 |
| NAT | -5.51 | -35.26 | Farol De Mae Luiza Lighthouse | NOAA | flask | Conway et al. 1994 |
| NMB | -23.58 | 15.03 | Gobabeb | NOAA | flask | Conway et al. 1994 |
| NWR | 40.05 | -105.59 | Niwot Ridge | NOAA | flask | Conway et al. 1994 |
| NWR | 40.05 | -105.59 | Niwot Ridge | NCAR | insitu | Stephens et al. 2011 |
| OTA | -38.52 | 142.82 | Otway | CSIRO | flask | Francey et al. 2003 |
| OXK | 50.03 | 11.81 | Ochsenkopf | NOAA | flask | Conway et al. 1994 |
| PAL | 67.97 | 24.12 | Pallas-Sammaltunturi | NOAA | flask | Conway et al. 1994 |
| PAL | 67.97 | 24.12 | Pallas-Sammaltunturi | FMI | insitu | Hatakka et al. 2003 |
| POC | | | Pacific Ocean Cruise | NOAA | flask | Conway et al. 1994 |
| PSA | -64.92 | -64 | Palmer Station | NOAA | flask | Conway et al. 1994 |
| RPB | 13.16 | -59.43 | Ragged Point | NOAA | flask | Conway et al. 1994 |
| RYO | 39.03 | 141.82 | Ryori | JMA | insitu | Tsutsumi et al. 2005 |
| SCT | 33.41 | -81.83 | Beech Island | NOAA | insitu | Andrews et al. 2014 |

| SEY | -4.68 | 55.53 | Mahe Island | NOAA | flask | Conway et al. 1994 |
|-----|-------|-------|-------------|------|-------|--------------------|
| SGP | 36.8 | -97.5 | Southern Great Plains | NOAA | flask | Conway et al. 1994 |
| SHM | 52.72 | 174.1 | Shemya Island | NOAA | flask | Conway et al. 1994 |
| SMO | -14.25 | -170.56 | Tutuila | NOAA | flask | Conway et al. 1994 |
| SMO | -14.25 | -170.56 | Tutuila | NOAA | insitu | Halter et al. 1988 |
| SNP | 38.62 | -78.35 | Shenandoah National Park | NOAA | insitu | Andrews et al. 2014 |
| SPL | 40.45 | -106.73 | Storm Peak Laboratory (Desert Research Institute) | NCAR | insitu | Stephens et al. 2011 |
| SPO | -89.98 | -24.8 | South Pole | NOAA | flask | Conway et al. 1994 |
| SPO | -89.98 | -24.8 | South Pole | NOAA | insitu | Gillette et al. 1987 |
| STR | 37.76 | -122.45 | Sutro Tower | NOAA | flask | Andrews et al. 2014 |
| SUM | 72.6 | -38.42 | Summit | NOAA | flask | Conway et al. 1994 |
| SYO | -69.01 | 39.59 | Syowa Station | NOAA | insitu | Morimoto et al. 2003 |
| TAP | 36.73 | 126.13 | Tae-ahn Peninsula | NOAA | flask | Conway et al. 1994 |
| THD | 41.05 | -124.15 | Trinidad Head | NOAA | flask | Conway et al. 1994 |
| USH | -54.85 | -68.31 | Ushuaia | NOAA | flask | Conway et al. 1994 |
| UTA | 39.9 | -113.72 | Wendover | NOAA | flask | Conway et al. 1994 |
| UUM | 44.45 | 111.1 | Ulaan Uul | NOAA | flask | Conway et al. 1994 |
| WAO | 52.95 | 1.12 | Weybourne | UEA | insitu | Forster and Bandy, 2006 |
| WBI | 41.73 | -91.35 | West Branch | NOAA | insitu | Andrews et al. 2014 |
| WGC | 38.27 | -121.49 | Walnut Grove | NOAA | insitu | Andrews et al. 2014 |
| WIS | 30.86 | 34.78 | Weizmann Institute of Science | NOAA | flask | Conway et al. 1994 |
| WKT | 31.32 | -97.33 | Moody | NOAA | insitu | Andrews et al. 2014 |
| WLG | 36.29 | 100.9 | Mt. Waliguan | NOAA | flask | Conway et al. 1994 |
| WSA | 43.93 | -60.02 | Sable Island | EC | insitu | Worthy et al. 2003 |
| YON | 24.47 | 123.02 | Yonagunijima | JMA | insitu | Tsutsumi et al. 2005 |
| ZEP | 78.91 | 11.89 | Ny-Alesund | NOAA | flask | Conway et al. 1994 |

Table A2. Validation sites. Aircraft data collected by NOAA/ESRL (Sweeney et al., 2015) and NIES (Machida et al., 2008)

| Site ID | Lat. | Lon. | Site/project name | Territory | Lab name |
|---------|------|------|-------------------|-----------|----------|
| ACG | 68 | -165 | Alaska Coast Guard | Alaska | NOAA |
| CAR | 41 | -104 | Briggsdale | Colorado | NOAA |
| CMA | 39 | -74 | Offshore Cape May | New Jersey | NOAA |
| CON | | | CONTRAIL | West Pacific | NIES |
| DND | 47 | -99 | Dahlen | North Dakota | NOAA |
| ESP | 49 | -127 | Estevan Point | British Columbia | NOAA |
| ETL | 54 | -105 | East Trout Lake | Saskatchewan | NOAA |
| INX | 40 | -86 | Indianapolis Flux Experiment | Indianapolis | NOAA |
| LEF | 46 | -90 | Park Falls | Wisconsin | NOAA |
| HIL | 40 | -88 | Homer | Illinois | NOAA |
| NHA | 43 | -71 | Offshore Portsmouth | New Hampshire | NOAA |
| PFA | 65 | -148 | Poker Flat | Alaska | NOAA |
| RTA | -21 | -160 | Rarotonga | Rarotonga | NOAA |
| SCA | 33 | -79 | Offshore Charleston | South Carolina | NOAA |
| SGP | 37 | -98 | Southern Great Plains | Oklahoma | NOAA |
| TGC | 28 | -97 | Offshore Corpus Christi | Texas | NOAA |

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

**Figures**

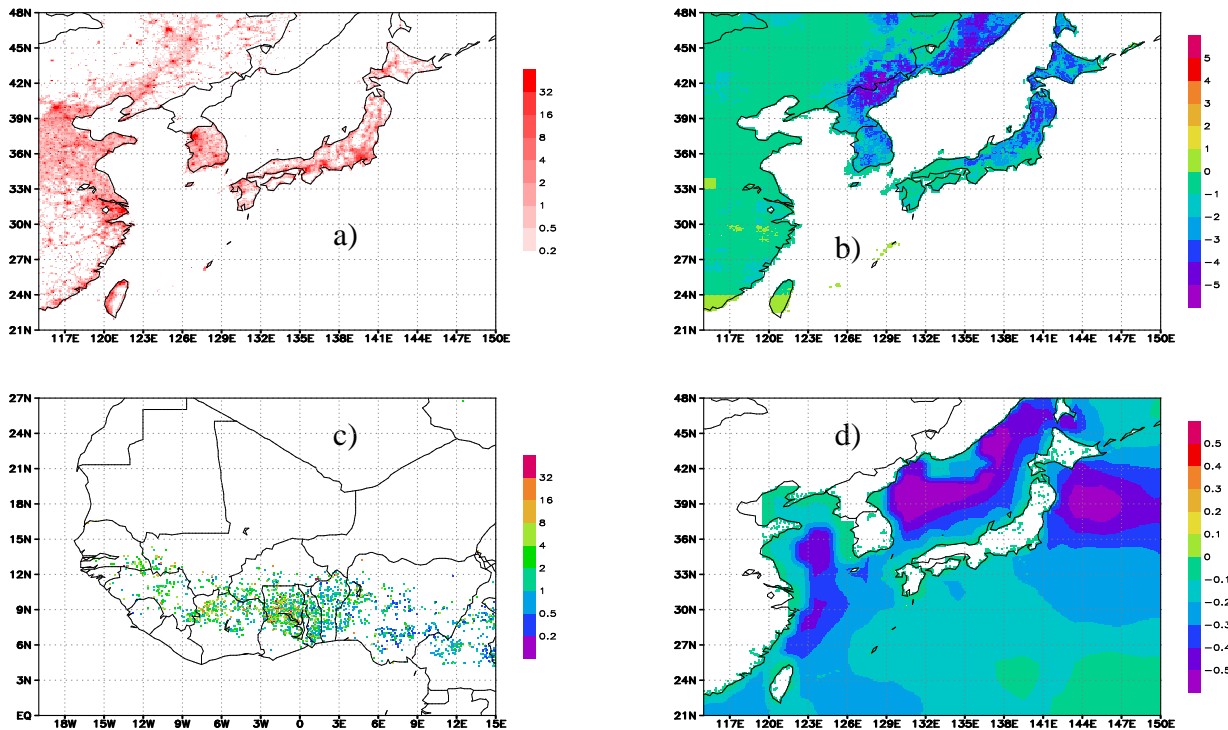

Figure 1: Examples of prior CO$_2$ fluxes (unit: gCm$^{-2}$day$^{-1}$): a) emissions from fossil fuel burning by ODIAC (Jan 2011), b) fluxes from terrestrial biosphere by optimized VISIT model (day 160, Jun 9), c) emissions from biomass burning by GFAS in Africa (Jan 10, 2011) and d) fluxes due to ocean-atmosphere exchange by the OTTM assimilation model (Jan 2011).

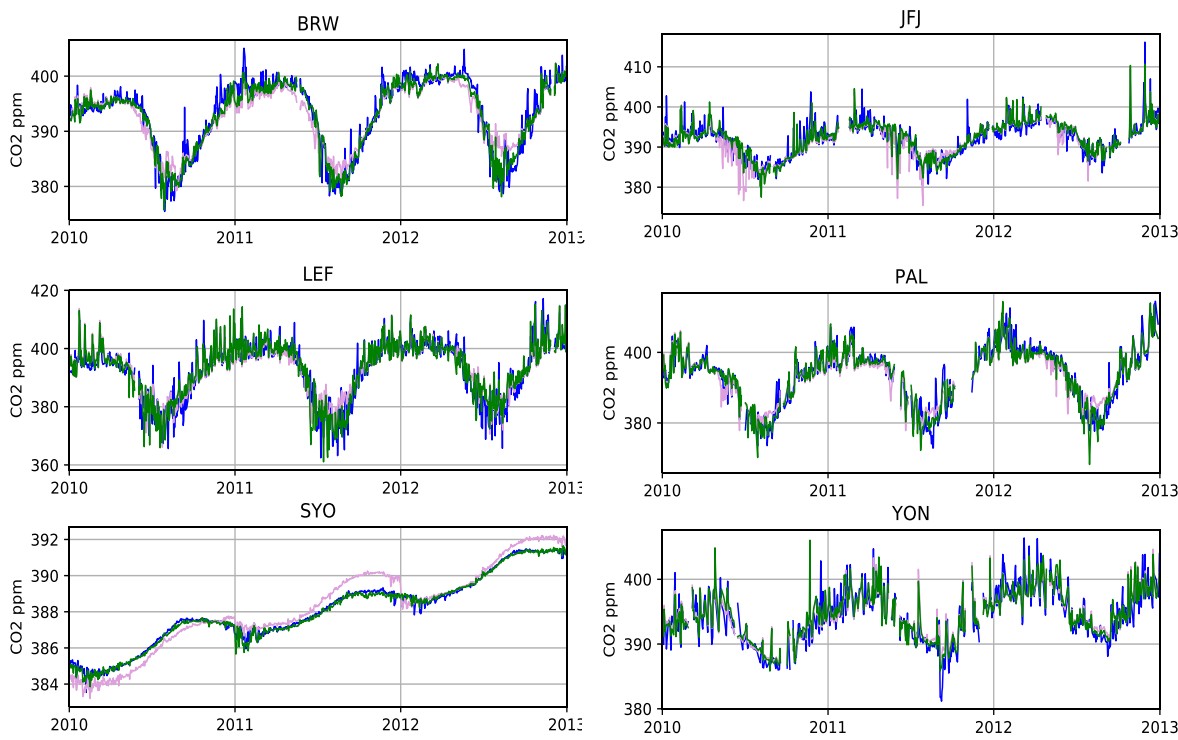

**Figure 2: Time series of simulated and observed concentrations (blue - observed, plum -forward (unoptimized), green – optimized) at Barrow (BRW), Jungfraujoch (JFJ), Wisconsin (LEF), Pallas (PAL), Syowa (SYO), and Yonagunijima (YON).**

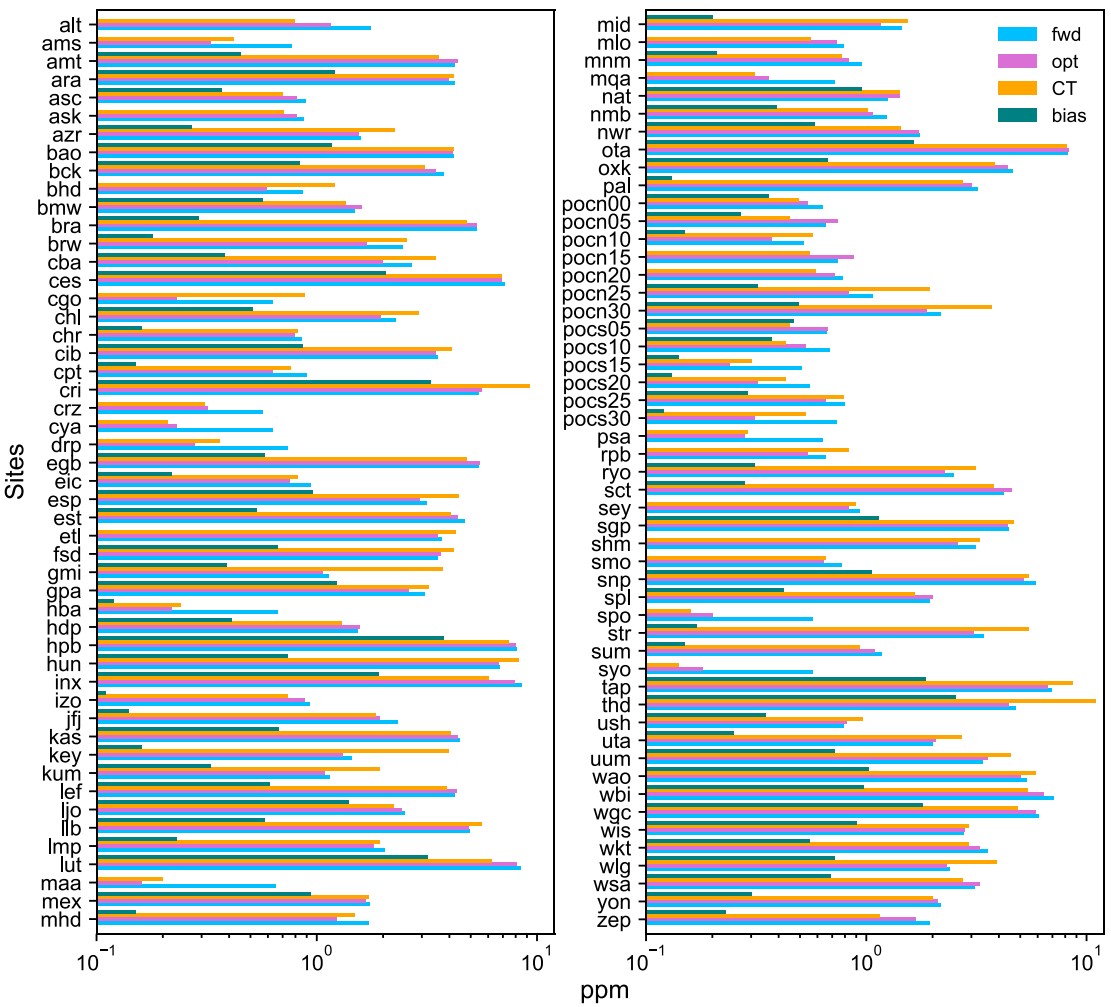

**Figure 3: RMS difference between model and observations and absolute bias in 2010-2012 for (surface) sites included in inversion (blue – prior, pink – optimized, orange – CT2017, green – absolute value of mean difference (bias) for optimized)**

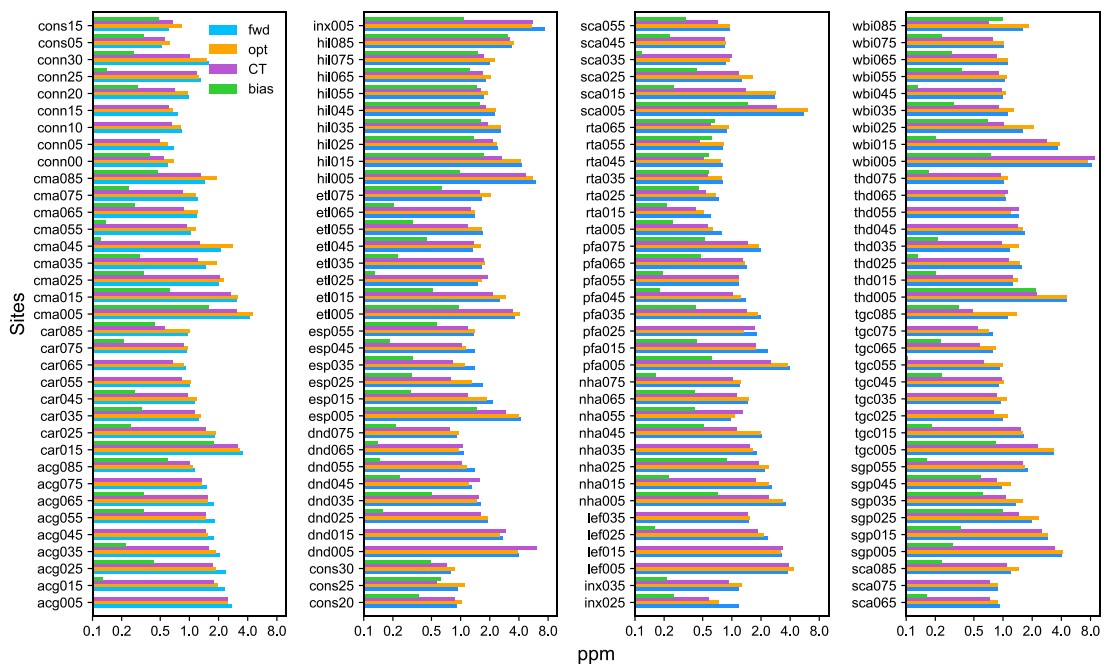

**Figure 4: RMS difference between model and observations and absolute bias in 2010-2012 for aircraft sites, not included in inversion (blue – prior, orange – optimized, magenta – CT2017, green – absolute value of mean difference (bias) for optimized)**

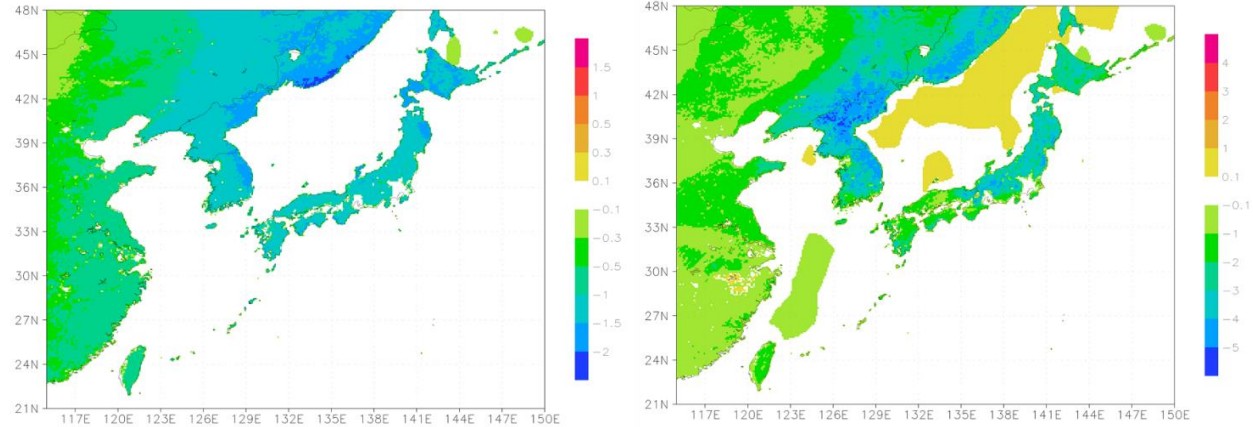

**Figure 5: Optimized flux correction (left) and posterior flux (right) maps for August 2011 (units gCm⁻²day⁻¹, fossil emissions excluded)**

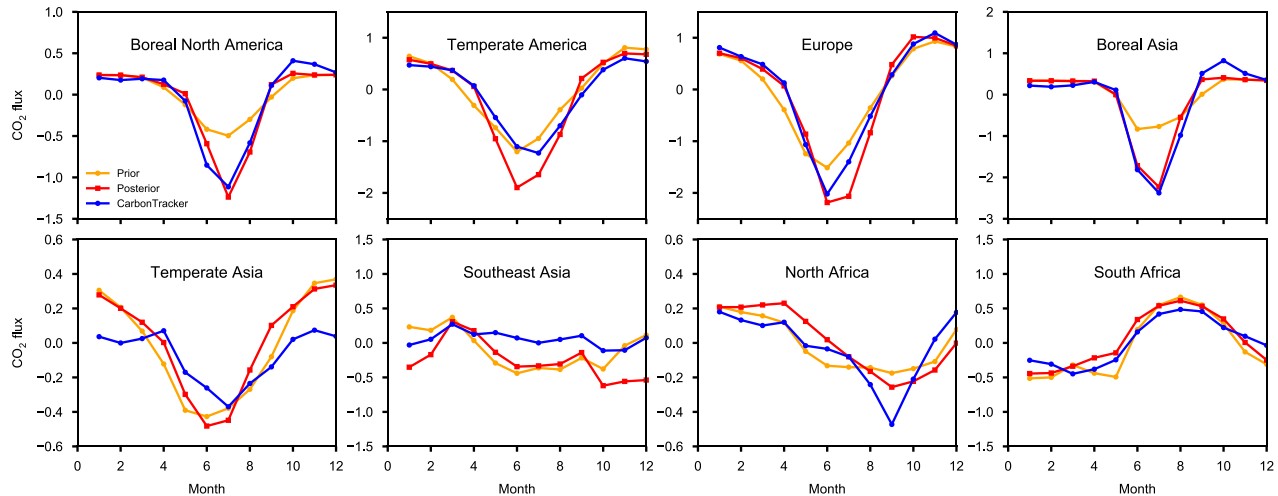

**Figure 6: Monthly mean prior, optimized and CarbonTracker fluxes (fossil emissions excluded), averaged for 2010-2012 and for selected Transcom-3 regions (units gCm⁻²day⁻¹).**

**Appendix figures.**

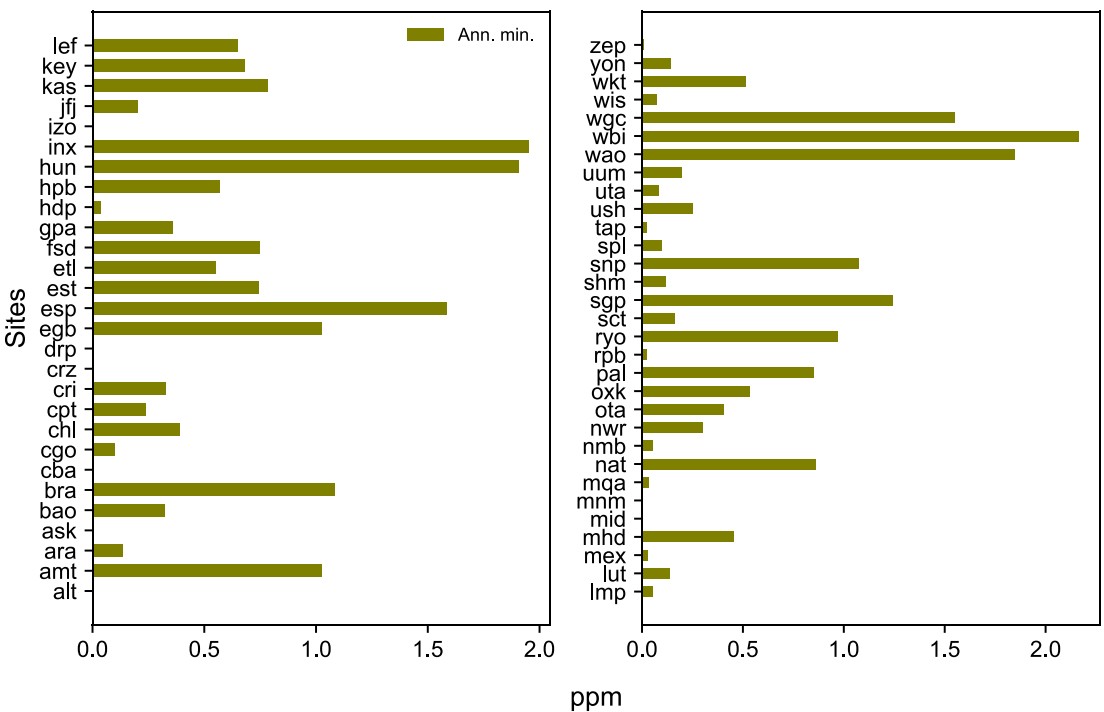

**Figure A1: Simulated diurnal cycle bias, 3-month mean in the growing season (units ppm).**

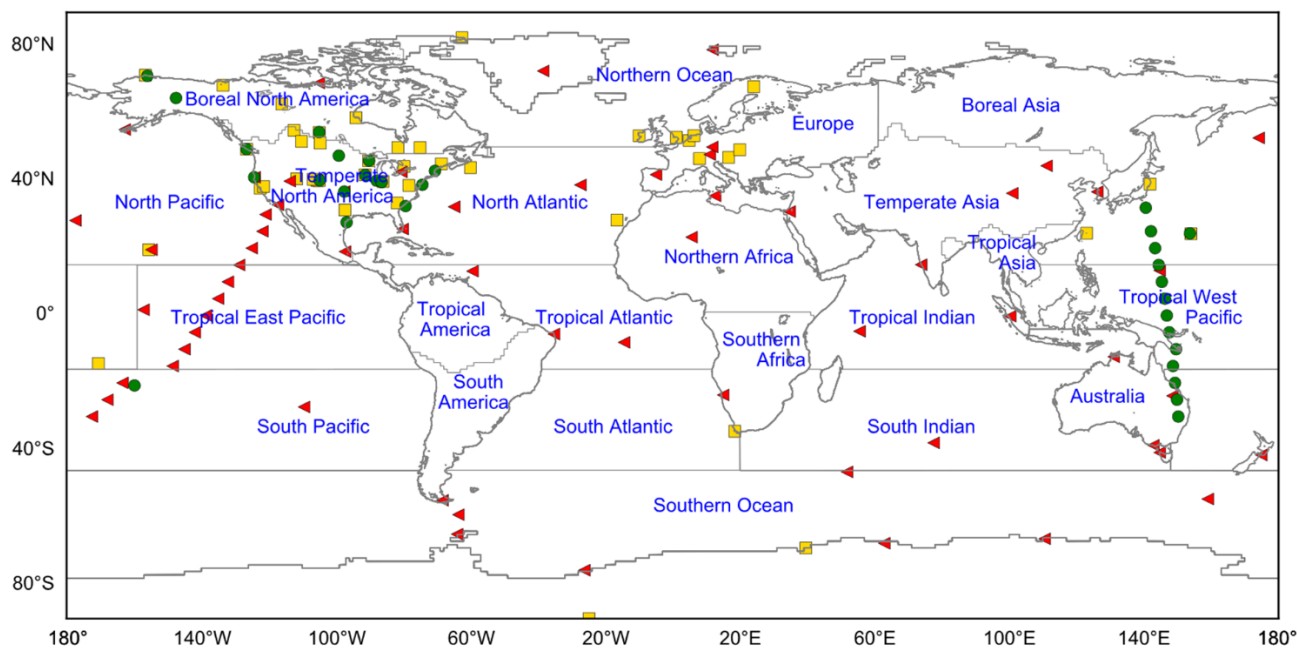

**Figure A2: A map of observation sites and Transcom regions (triangles - surface flask sites, squares -continuous, circles – aircraft) used in this study.**

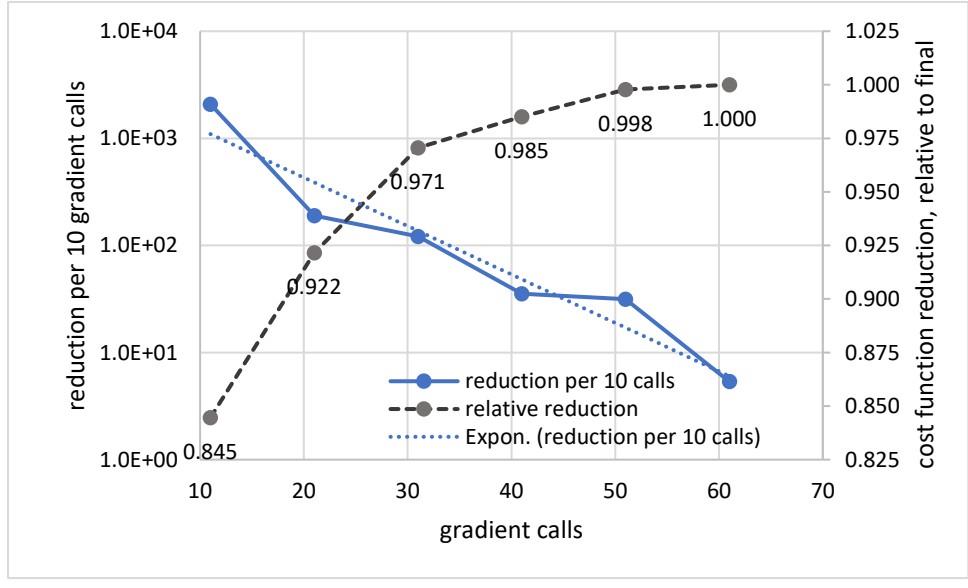

Figure A3: Rate of cost function decline with gradient calls for the extended year 2011 inversion and reduction relative to 61 gradient calls.