# Peer review of "Technical note: A high-resolution inverse modelling technique for estimating surface CO2 fluxes based on the NIES-TM - FLEXPART coupled transport model and its adjoint."

_Atmospheric Chemistry and Physics, 2020_

## Referee Comment (RC1) · Anonymous Referee #1 · 29 Jun 2020

The paper is a technical note, reporting on minor improvements in the setup already described in Belikov et al., 2016. The improvements seem to be exclusively technical (some improvements in the memory management, and the use of a different tool to derive the adjoint code, which would fit better in GMD than in ACP). Furthermore there is not demonstration that it achieves any better results of performance compared to that setup or to comparable inverse models. In fact, the only results presented are a series of model-data mismatches, which do not demonstrate much, beyond the fact that the model is indeed able to improve the fit to observations (the contrary would be very

worrying!). Finally, I don't think that the setup is adequate for what it aims to achieve (it makes no sense to optimize fluxes at a 0.1° resolution with covariance lengths of 500 km). For these reasons, I unfortunately, cannot recommend that paper for further publication in ACP.

**1  Major comments:**

- The modeling setup is an evolution of the one used in Belikov et al., 2016. An ACP technical note should summarize "new developments, significant advances, or novel aspects of experimental and theoretical methods and techniques that are relevant for scientific investigations within the scope of the journal" (https://www.atmospheric-chemistry-and-physics.net/about/manuscript_types.html). If the developments introduced represent such a significant advance, I think it should be demonstrated (at least by a comparison with the old setup). The only results that are presented are model-observation mismatches (with assimilated and validation data), compared with those achieved by CarbonTracker. It's a useful diagnostic, but definitely not a result, and not a proof that an inversion performs better than another one (overfitting the observations with unrealistic flux adjustments is possible). If the improvements are limited to performance, then not only this should be more explicitly mentioned, but also I think ACP is not the good journal for this.

- I don't understand the possible interest of optimizing fluxes at a 0.1° resolution when 500 km correlation distances are used. With these correlations the flux adjustments patterns will span several hundreds of contiguous grid cells (500 km is almost 5° at the equator): this is just a wast of resources, the same flux adjustments can be achieved with a lower resolution inversion and proper accounting of the model representation error (or at least that's my intuition. I would be happy

to be proven wrong, but nothing in the paper does that). So the whole setup seems to just add a layer of complexity (and potentially of biases), without clear performance or accuracy advantages. I could the need for NIES of an inversion system capable of assimilating satellite data at a high resolution, but this is not what is presented here. There is no guarantee that the inversion would still be feasible with shorter correlation length and a lot more data.

**2 Minor comments:**

p3, l9: A disadvantage ... , is addressed ... aggregated flux regions: there is something missing in that sentence

p3, l 15: thus $\rightarrow$ this

p6, l26: isn't it dangerous to base the uncertainty on GPP? (what about winter time, when GPP is near zero ... there would still be uncertainty on the respiration term ...

p8: If x is the optimized flux (as stated in line 6), then the minimum of the second term of Equation 1 is obtained for $||x|| = 0$ (the flux is minimized, not its distance to its prior). Or "x" is a flux correction, but then it is the first term of the equation that is wrong, Either way, I don't think that it's what the authors meant ...

p9, l20: isn't it a problem to have all footprints stopping at 0 GMT, regardless of when they started? That means that some footprints will systematically span longer (I guess up to 4 days?) depending on their origin longitude?

p11, l16: What is the "implicit diffusion with directional splitting"? I think it's technical enough to be worth a more detailed explanation. The rest of the paragraph is dedicated to explaining the merits of that diffusion algorithm, so I assume it's an important part of the setup, but if I wanted to reproduce it, I would have no idea how to do it (based on what's written in the paper).

p12, l19: "our report is limited to technical development" ==> I don't think that it's the aim of ACP then . . .

p12, l23: Three-month → Three-months

p14, l23: the spatial resolution is roughly 500 km (i.e. the length of the prescribed correlations), not 0.1°. The system might technically be ready for inversions at 0.1°, but given the way it is setup, I doubt that the optimized fluxes would be any different if the optimization was done at a 1° or 2° resolution. Of course, I would be happy to be proven wrong, but the authors don't even try . . .

---

## Referee Comment (RC2) · Anonymous Referee #2 · 30 Jun 2020

The 'Technical note' by Maksyutov and co-authors describes the technical details of a CO2 flux inversion technique based on the coupled Eulerian-Lagrangian transport model NIES-TM-FLEXPART. The coupled system operates at a high spatial resolution of CO2 fluxes of 0.1° x 0.1° globally and also attempts a flux inversion at this resolution. As such the approach is novel and promising. The paper is well structured and written. The performance of the inversion is documented by time series comparisons/evaluations of different data sources assimilated and not assimilated by the system and a comparison to another independent inversion system. However, the inverted flux fields are never shown/discussed in the manuscript, which makes it rather difficult to judge if the inversion yielded reasonable results. Even though the manuscript is a 'technical note', it would be very beneficial to overcome this shortcoming (detailed suggestion below) before publication.

Major comments

Section 5: Evaluating the performance of an inverse modelling system is not straight-forward. Restricting this evaluation to a comparison of model skill for the assimilated and additional independent concentration time series is not sufficient. By over-fitting the flux fields a very good agreement of the posterior concentrations with the observations may be achieved but the flux fields may contain unrealistic detail in order to achieve this. Given the large degree of freedom in the fluxes, as indicated by the fact that grid and time resolved fluxes are inverted from a relatively limited set of observations, there seems to be a high risk for the presented inversion setup to over-fit the solution. Since no flux fields are presented, it is impossible to judge this possibility. Therefore, I would encourage the authors to extend their discussion of results in section 5 to include a brief analysis of the obtained flux fields. I can see that the authors have planed this for a later publication and, hence, I don't think this needs to be very quantitative here, but the presented flux fields should document the validity of the inversion approach. A qualitative comparison with flux fields obtained from CarbonTracker (as done for the concentration time series) would also be beneficial.

Minor comments

P3,L18: Resolution of coupled Eulerian-Lagrangian models. I find it a bit misleading to say that the transport in these models is run at a resolution of (as in the cited publication) 1 km. Yes, technically the transport is not run at any fixed resolution in the Lagrangian sense, but the driving meteorology is still determining what scales of motion can be correctly resolved by the model. The Lagrangian model may still have some skill in the sub-resolved range, but basically it degenerates to a Gaussian plume

at these scales with constant wind direction, speed and dispersion characteristics. This fact is not sufficiently highlighted throughout the manuscript. Another example is P4, where the transport system for the current study is introduced. The driving meteorology is $1.25°x1.25°$ this is certainly not sufficient for a detailed transport description in complex, mountainous terrain, but also not for coastal areas. Since sites from both environments are contained in the list of sites used for assimilation, this limitation should be discussed in more detail. Next to spatial resolution also temporal resolution is important for a transport description in the mentioned environments. Although in the following, the use of observations was restricted to certain times of the day, these observations still carry the transport history for a longer period and if temporal variability in the transport is not sufficiently described may lead to biases in the simulated concentrations as well.

P5,L13f: If I understand this correctly, there is no diurnal cycle of the biospheric $CO_2$ flux considered in the model setup. How much is this simplification limiting the model performance? Not all sites used for the inversion are remote coastal sites but are surrounded by dense vegetation. How much does the constant diurnal flux and the restriction to afternoon observations introduce a bias in the flux inversion? In general, I have the feeling that the low temporal resolution of fluxes does not keep up with their high spatial resolution in the current setup.

P6,L8: How much do the posteriori fluxes actually depend on the chosen biospheric flux climatology? Given the large year-to-year variability in biospheric fluxes, is it sufficient to operate with a climatology of prior fluxes? Was this evaluated by choosing a different averaging interval or even individual year for the prior climatology.

Section 3.3: Are all biomass burning emissions considered to be released at the model surface or was any kind of vertical emission profile used? Again, this may be crucial when considering transport simulations at the mesoscale.

P7, L9 and P9, L21: According to the first text location only afternoon samples were

considered for the inversion. However, according to the second location daily average footprints of the Lagrangian model were applied. If that is really the case the footprints are not representative for the used observations, largely neglecting temporal variability once again!

P9, L25ff: The description of the forward model steps. From this description it is not clear to me how the concentration increments from the different models are added to avoid double counting of the fluxes (step 2.c). Shouldn't the Eulerian model use different fluxes than the Lagrangian (cropping those that are covered by the Lagrangian model)?

P10, L24: Initial conditions are used from an optimised run from the previous year. But then the question remains how the previous year was initialised. Was this done with a spin-up run?

Section 5.1: Why is RMSE used as the sole estimator of model performance? RMSE will decrease even if only the baseline fits better after optimisation. Most of the regional flux information, however, is stored in the peak concentrations, for which a more robust performance estimator could be a bias corrected RMSE or the coefficient of determination. The bias should be reported as well. Taylor skill score could be another performance parameter that would be more suited to focus more on the short term variability.

Sites: It would be useful to see the sites and the aircraft locations on a map. Would help to judge which areas are not well covered by assimilated observations in comparison to validation data. Such a plot should also contain the information of flask vs. continuous observations.

Technical comments

P3,L10: Start a new sentence after '... Kaminski et al. (2001). This is addressed by ..."

P12,L14: 'previous' instead of 'pervious'.

Figure 1, caption: Label as 'Examples of ...'

Figure 3+4: Since the x-axis is not along a continuous variable, I would suggest to not include lines in the plot or even use a barplot instead of symbols. The lines are just confusing and have no physical meaning.

―――――――――――――――――――

---

## Short Comment (SC1) · 30 Jun 2020

A short comment by S. Maksyutov (first author) in reply to Anonymous referee #1 comment posted June 29, 2020

The review comment states: "The paper is a technical note, reporting on minor improvements in the setup already described in Belikov et al., 2016. The improvements seem to be exclusively technical (some improvements in the memory management, and the use of a different tool to derive the adjoint code, which would fit better in GMD

than in ACP). Furthermore there is not demonstration that it achieves any better results of performance compared to that setup or to comparable inverse models."

Author's reply: There is some misunderstanding about developments made since the mentioned paper. It should be noted that in a paper by Belikov et al., (2016), there was no attempt to do the inversion, instead, it focused on development of forward coupled model (at lower resolution of 1 degree), its adjoint, the adjoint accuracy and performance.

In this study, (1) the Lagrangian model resolution was increased to 0.1 degree, and necessary prior fluxes were developed; (2) Flux covariance operator was developed specifically to handle the challenges of operation at high spatial resolution; (3) Iterative optimization technique was implemented and multiple (time consuming) inversion trials were performed before achieving reported results.

The review comment: "In fact, the only results presented are a series of model-data mismatches, which do not demonstrate much, beyond the fact that the model is indeed able to improve the fit to observations (the contrary would be very worrying!). "

Author's reply: Still, do demonstrate that the technical development is valid, and the inverse model does work, showing the fit to the observations is desirable.

The review comment: "Finally, I don't think that the setup is adequate for what it aims to achieve (it makes no sense to optimize fluxes at a 0.1° resolution with covariance lengths of 500 km)."

Author's reply: Using the same resolution in inversion as in transport is achieved in our case with a minor additional computational cost (due to efficient covariance operator). The covariance scale is a tunable parameter, it can be set according to information content available in the observations. Many inverse modeling studies (eg Chevallier et al, 2010) do not assume the current observing network provides enough information to constrain the land biosphere fluxes globally at a higher resolution than 500 km. It

is mentioned (Chevallier et al, 2010) that with shorter covariance scales the model may take more iterations to converge. Accordingly, the transport model resolution is often higher here than the effective resolution of the inverse model. The rationale for using higher resolution inversion in comparison to lower resolution, such as using large regions, is to reduce aggregation error (Kaminsky et al., 2001).

References:

Belikov, D. A., Maksyutov, S., Yaremchuk, A., Ganshin, A., Kaminski, T., Blessing, S., Sasakawa, M., Gomez-Pelaez, A. J., and Starchenko, A.: Adjoint of the global Eulerian-Lagrangian coupled atmospheric transport model (A-GELCA v1.0): development and validation, Geoscientific Model Development, 9, 749-764, 10.5194/gmd-9-749-2016, 2016.

Chevallier, F., et al.: CO2 surface fluxes at grid point scale estimated from a global 21 year reanalysis of atmospheric measurements, J. Geophys. Res., 115, D21307, doi:10.1029/2010JD013887, 2010.

Kaminski, T., Rayner, P. J., Heimann, M., and Enting, I. G. : On aggregation errors in atmospheric transport inversions, J. Geophys. Res., 106( D5), 4703– 4715, doi:10.1029/2000JD900581, 2001.
* * *

---

## Referee Comment (RC3) · Anonymous Referee #1 · 3 Aug 2020

I initially suggested rejecting the paper, since I thought it was just a minor evolution of a previous work from the same team (e.g. Belikov et al., 2016) and because of the insufficient quality (and quantity!) of the results presented. The authors have clarified in a reply that the difference with Belikov et al., 2016 was larger than what I had understood. I thank them for this clarification and apologize for the confusion.

Even though the sections 1 to 4 (introduction and methods) are well written, it remains difficult to give good gradings to paper (especially for the "scientific quality" and "presentation quality" criterias) as the results presented are absolutely insufficient to prove

that the model is working as expected, and more generally to support some of the main claims of the conclusions (p15, l5-7). I am however willing to change my recommendation to major revisions, in the light of the clarifications provided by the authors.

**Detailed reply to author's comment**

The review comment states: "The paper is a technical note, reporting on minor improvements in the setup already described in Belikov et al., 2016. The improvements seem to be exclusively technical (some improvements in the memory management,and the use of a different tool to derive the adjoint code, which would fit better in GMD than in ACP). Furthermore there is not demonstration that it achieves any better results of performance compared to that setup or to comparable inverse models."

Author's reply: There is some misunderstanding about developments made since the mentioned paper. It should be noted that in a paper by Belikov et al., (2016), there was no attempt to do the inversion, instead, it focused on development of forward coupled model (at lower resolution of 1 degree), its adjoint, the adjoint accuracy and performance. In this study, (1) the Lagrangian model resolution was increased to 0.1 degree, and necessary prior fluxes were developed; (2) Flux covariance operator was developed specifically to handle the challenges of operation at high spatial resolution; (3) Iterative optimization technique was implemented and multiple (time consuming) inversion trials were performed before achieving reported results.

Ok, noted. But then I would suggest to make this much clearer in the paper. The paragraph starting at line 15 on page 3 is particularly confusing.

[Figure]

The review comment: "In fact, the only results presented are a series of model-data mismatches, which do not demonstrate much, beyond the fact that the model is indeed able to improve the fit to observations (the contrary would be very worrying!). "Author's reply: Still, do demonstrate that the technical development is valid, and the inverse model does work, showing the fit to the observations is desirable.

What is your criteria to say that "the inverse model does work"? If the criteria is just to "improve the fit to the observations", then indeed it works, and you demonstrate it. But the aim of doing an inversion is to find the optimal value for the optimized parameters, in you case the CO2 fluxes (given the information from the prior and from the observations), which you formalize as finding the vector that minimizes J(x) in Equation 1:

1. How can I be sure that the posterior fluxes indeed minimize J(x)? There are certainly vectors that improve the fit to the observations but increase the value of J, which your system could find if it malfunctions). Even if it works properly, how do you know that 45 iterations is enough?

2. And how can I be sure that the value of x that minimizes J is indeed closer to reality than the prior? (if your transport model is strongly biased, or if the uncertainties matrices are not adapted, you can totally end up with posterior fluxes that are worse than the prior).

Even if you can't provide a formal answer to these points, you can make steps towards addressing them! And at the very least show the posterior fluxes and how they differ from the prior!

The review comment: "Finally, I don't think that the setup is adequate for what it aims to achieve (it makes no sense to optimize fluxes at a 0.1°

resolution with covariance lengths of 500 km)."Author's reply: Using the
same resolution in inversion as in transport is achieved in our case with a
minor additional computational cost (due to efficient covariance operator).

Could you be more specific? What is the cost of a NIES-TM-FLEXPART inversion,
compared to a NIES-TM inversion? And how different are the posterior fluxes (and
how better are they)?

> The covariance scale is a tunable parameter, it can be set according to
> information content available in the observations. Many inverse modeling
> studies (eg Chevallier et al, 2010) do not assume the current observing
> network provides enough information to constrain the land biosphere fluxes
> globally at a higher resolution than 500 km. It is mentioned (Chevallier
> et al, 2010) that with shorter covariance scales the model may take more
> iterations to converge. Accordingly, the transport model resolution is often
> higher here than the effective resolution of the inverse model. The rationale
> for using higher resolution inversion in comparison to lower resolution, such
> as using large regions, is to reduce aggregation error (Kaminsky et al.,
> 2001).

I am not questioning the benefit of solving the fluxes at a higher resolution, just the ad-
equacy of that combination of covariance scales and transport/optimization resolution.
If your observations limit you to an effective resolution of 500 km, then I think that you
don't need such a high resolution transport model. And you had enough observations
to require such a high resolution transport (if you have enough obs. to use covariance
scales of e.g. 50 km), would your inversions still be feasible (it would require much
more iterations).

If you can demonstrate that you get much better results (fluxes, not just observation
fits!) with 500km/0.1° than with e.g. 500km/1°, then it could be a strong incentive from

other groups to implement a similar coupling, so I think that it would be really interesting to do that comparison properly.

---

## Author Comment (AC1) · 15 Sep 2020

Authors comments on 'Technical note: A high-resolution inverse modelling technique for estimating surface CO2 fluxes based on the NIES-TM – FLEXPART coupled transport model and its adjoint'

Authors are grateful to reviewers for work they have done reading the paper and preparing the comments. The comments were very useful for realizing the deficiencies in the manuscript and the presented study and will serve as expert guidelines for further improvements.

We prepared revisions to the manuscript reflecting the review comments, as summarized below. Original review comments appear in black, and replies in color.

Replies to comments by Anonymous reviewer #1

RC1

The paper is a technical note, reporting on minor improvements in the setup already described in Belikov et al., 2016. The improvements seem to be exclusively technical (some improvements in the memory management, and the use of a different tool to derive the adjoint code, which would fit better in GMD than in ACP). Furthermore there is not demonstration that it achieves any better results of performance compared to that setup or to comparable inverse models. In fact, the only results presented are a series of model-data mismatches, which do not demonstrate much, beyond the fact that the model is indeed able to improve the fit to observations (the contrary would be very worrying!). Finally, I don't think that the setup is adequate for what it aims to achieve (it makes no sense to optimize fluxes at a 0.1° resolution with covariance lengths of 500 km). For these reasons, I unfortunately, cannot recommend that paper for further publication in ACP.

1 Major comments:
• The modeling setup is an evolution of the one used in Belikov et
al., 2016. An ACP technical note should summarize "new developments, significant advances, or novel aspects of experimental and theoretical methods and techniques that are relevant for scientific investigations within the scope of the journal" (https://www.atmospheric-chemistry-andphysics.net/about/manuscript_types.html). If the developments introduced represent such a significant advance, I think it should be demonstrated (at least by a comparison with the old setup). The only results that are presented are model observation mismatches (with assimilated and validation data), compared with those achieved by CarbonTracker. It's a useful diagnostic, but definitely not a result, and not a proof that an inversion performs better than another one (overfitting the observations with unrealistic flux adjustments is possible). If the improvements are limited to performance, then not only this should be more explicitly mentioned, but also I think ACP is not the good journal for this.

Response: It is true that we present technical development, which is still not in perfect shape and needs more improvements and tuning, constructed amid limited supply of the

necessary components. But, on the way we developed several techniques that can be of interest to readers. To make the study more complete we add section on optimized fluxes in the revised manuscript.

• I don't understand the possible interest of optimizing fluxes at a 0.1◦
Resolution when 500 km correlation distances are used. With these correlations the flux adjustments patterns will span several hundreds of contiguous grid cells (500 km is almost 5◦ at the equator): this is just a wast of resources, the same flux adjustments can be achieved with a lower resolution inversion and proper accounting of the model representation error (or at least that's my intuition. I would be happy to be proven wrong, but nothing in the paper does that). So the whole setup seems to just add a layer of complexity (and potentially of biases), without clear performance or accuracy advantages. It could the need for NIES of an inversion system capable of assimilating satellite data at a high resolution, but this is not what is presented here. There is no guarantee that the inversion would still be feasible with shorter correlation length and a lot more data.

Reply: It should be noted that the flux corrections are not smoothed with 500 km scale filter. The correlations are applied to a smooth field of a scaling factor which is multiplied by high-resolution flux uncertainty field to give flux corrections, which is explained in more detail in the text added to sect 4.2.

"This design of covariance operator helps preserving high resolution structure of the resultant flux corrections, given by $x = L \cdot z = u_F \cdot (L_{xy} \otimes L_t) \cdot m \cdot z$ , as it can be factored into multiple of uncertainty $u_F$ and scaling factor $S = (L_{xy} \otimes L_t) \cdot m \cdot z$ as $x = u_F \cdot S$ . While the scaling factor $S$ is smoothed with covariance length of 500 km, the original structure of spatial heterogeneity of surface flux uncertainty $u_F$ is still present at original high resolution in the optimized flux corrections $x$. "
With given resolution of fluxes and transport model of 0.1 degree, inversion can be done at various settings independently from resolution of transport model, prior fluxes and flux uncertainties, such as using 22 regions globally, 1x1 degree regions or 0.1 degree grid. The reason we do inversion at the same resolution as flux dimension is technical, as it is simpler to implement.

Minor comments:
p3, l9: A disadvantage . . . , is addressed . . . aggregated flux regions: there is something missing in that sentence

Corrected, by starting new sentence.

p3, l 15: thus → this

Corrected.

p6, l26: isn't it dangerous to base the uncertainty on GPP? (what about winter time, when GPP is near zero . . . there would still be uncertainty on the respiration term . . .

Reply. That is useful notice, the respiration field would serve as better base for biospheric flux uncertainty, due to better seasonal coverage, but we did not have it at high resolution (now of course, a number of alternative datasets is made available by Jung et al. 2020, Jones et al. 2016)

p8: If x is the optimized flux (as stated in line 6), then the minimum of the second term of Equation 1 is obtained for $\|x\| = 0$ (the flux is minimized, not its distance to its prior). Or "x" is a flux correction, but then it is the first term of the equation that is wrong, Either way, I don't think that it's what the authors meant . . .

Reply: The paragraph is revised. The first term used in place of observations a residual misfit $r$ - difference between observations and simulation with prior fluxes, as was introduced on page 7, lines 22-26

p9, l20: isn't it a problem to have all footprints stopping at 0 GMT, regardless of when they started? That means that some footprints will systematically span longer (I guess up to 4 days?) depending on their origin longitude?

Reply: Even if some footprints take more time it is accounted for when the sensitivity is recorded, those with longer time will produce more signal. To clarify, added sentence: 'The coupling time is set to be within 2 to 3 days before observation event..'

p11, l16: What is the "implicit diffusion with directional splitting"? I think it's technical enough to be worth a more detailed explanation. The rest of the paragraph is dedicated to explaining the merits of that diffusion algorithm, so I assume it's an important part of the setup, but if I wanted to reproduce it, I would have no idea how to do it (based on what's written in the paper).

Reply: We add more text as explanation for use of diffusion to approximate covariance.

p12, l19: "our report is limited to technical development" ==> I don't think that it's the aim of ACP then . . .

Reply: Historically, there have been some publications on similar line in ACP, in a format of technical note.

p12, l23: Three-month → Three-months

Corrected

p14, l23: the spatial resolution is roughly 500 km (i.e. the length of the prescribed correlations), not 0.1◦ . The system might technically be ready for inversions at 0.1◦ , but given the way it is setup, I doubt that the optimized fluxes would be any different if the optimization was done at a 1◦ or 2◦ resolution. Of course, I would be happy to be proven wrong, but the authors don't even try . .

Reply. To counter impression that the optimized flux resolution could be 500 km, explanation was added on composition of posterior flux correction, in section 4.2 (prior covariance) and a figure in section 5.2 (posterior fluxes)

RC3

I initially suggested rejecting the paper, since I thought it was just a minor evolution of a previous work from the same team (e.g. Belikov et al., 2016) and because of the insufficient quality (and quantity!) of the results presented. The authors have clarified in a reply that the difference with Belikov et al., 2016 was larger than what I had understood. I thank them for this clarification and apologize for the confusion.
Even though the sections 1 to 4 (introduction and methods) are well written, it remains difficult to give good gradings to paper (especially for the "scientific quality" and "presentation quality" criterias) as the results presented are absolutely insufficient to prove that the model is working as expected, and more generally to support some of the main claims of the conclusions (p15, l5-7). I am however willing to change my recommendation to major revisions, in the light of the clarifications provided by the authors.

> "Detailed reply to author's comment
> The review comment states: "The paper is a technical note, reporting on minor improvements in the setup already described in Belikov et al., 2016. The improvements seem to be exclusively technical (some improvements in the memory management,and the use of a different tool to derive the adjoint code, which would fit better in GMD than in ACP). Furthermore there is not demonstration that it achieves any better results of performance compared to that setup or to comparable inverse models."
> Author's reply: There is some misunderstanding about developments made since the mentioned paper. It should be noted that in a paper by Belikov et al., (2016), there was no attempt to do the inversion, instead, it focused on development of forward coupled model (at lower resolution of 1 degree), its adjoint, the adjoint accuracy and performance. In this study, (1) the Lagrangian model resolution was increased to 0.1 degree, and necessary prior fluxes were developed; (2) Flux covariance operator was developed specifically to handle the challenges of operation at high spatial resolution; (3) Iterative optimization technique was implemented and multiple (time consuming) inversion trials were performed before achieving reported results."

Ok, noted. But then I would suggest to make this much clearer in the paper. The paragraph starting at line 15 on page 3 is particularly confusing.

Reply. The paragraph starting at line 15 on page 3 was revised.

> "The review comment: "In fact, the only results presented are a series of

model-data mismatches, which do not demonstrate much, beyond the fact that the model is indeed able to improve the fit to observations (the contrary would be very worrying!). "Author's reply: Still, do demonstrate that the technical development is valid, and the inverse model does work, showing the fit to the observations is desirable."

What is your criteria to say that "the inverse model does work"? If the criteria is just to "improve the fit to the observations", then indeed it works, and you demonstrate it. But the aim of doing an inversion is to find the optimal value for the optimized parameters, in you case the CO2 fluxes (given the information from the prior and from the observations), which you formalize as finding the vector that minimizes J(x) in Equation 1:

> 1. How can I be sure that the posterior fluxes indeed minimize J(x)? There are certainly vectors that improve the fit to the observations but increase the value of J, which your system could find if it malfunctions). Even if it works properly, how do you know that 45 iterations is enough?

Reply: To clarify, the comment that inversion works only relates to computational side not to the physical result for fluxes, the flux biases and biases in simulated concentrations. That should require an extra work to improve. Fortunately, we never see a cost function increasing in the output of M1QN3 minimizer. We added a text: Figure A3 in Appendix presents the cost function reduction in case of optimizing fluxes for 2011 and completing 61 gradient calls. The cost function reduction declines nearly exponentially, by almost 3 times for each 10 gradient calls completed. Relative improvement between 41 and 61 gradient calls is 1.5% of the total reduction from the first to the 61 gradient calls.

> 2. And how can I be sure that the value of x that minimizes J is indeed closer to reality than the prior? (if your transport model is strongly biased, or if the uncertainties matrices are not adapted, you can totally end up with posterior fluxes that are worse than the prior).

Could you be more specific? What is the cost of a NIES-TM-FLEXPART inversion, compared to a NIES-TM inversion? And how different are the posterior fluxes (and how better are they)?

Reply: We planned to do more improvements before producing the flux estimate that can be used for scientific objectives, this paper is meant to introduce a technique. A figure comparing prior, posterior and CarbonTracker was added. Added a relative CPU time information as: "The fraction of CPU time spent on running the Eulerian component of the coupled transport model is 82%, on the Lagrangian component 1%, and on covariance 17%."

"The covariance scale is a tunable parameter, it can be set according to information content available in the observations. Many inverse modeling studies (eg Chevallier et al, 2010) do not assume the current observing network provides enough information to constrain the land biosphere fluxes globally at a higher resolution than 500 km. It is mentioned (Chevallier et al, 2010) that with shorter covariance scales the model may take more iterations to converge. Accordingly, the transport model resolution is often higher here than the effective resolution of the inverse model. The rationale for using higher resolution inversion in comparison to lower resolution, such as using large regions, is to reduce aggregation error (Kaminsky et al., 2001)."

I am not questioning the benefit of solving the fluxes at a higher resolution, just the adequacy of that combination of covariance scales and transport/optimization resolution. If your observations limit you to an effective resolution of 500 km, then I think that you don't need such a high resolution transport model. And you had enough observations to require such a high resolution transport (if you have enough obs. to use covariance scales of e.g. 50 km), would your inversions still be feasible (it would require much more iterations). If you can demonstrate that you get much better results (fluxes, not just observation fits!) with 500km/0.1° than with e.g. 500km/1°, then it could be a strong incentive from other groups to implement a similar coupling, so I think that it would be really interesting to do that comparison properly.

Reply. It is possible that coupled 1 deg model will perform similarly to 0.1 deg model for estimating the land biosphere and ocean fluxes. However, the push for higher resolution is driven by a need to estimate the anthropogenic emissions. Our study is a step in that direction (as was mentioned in introduction, page 4 lines 6-7), and the high resolution has to be used there even when the land biosphere and ocean fluxes are better estimated with 1 degree resolution model. Thus, best direction for such objective is to keep improving the 0.1 deg setup, making effort to reduce the biases in the land biosphere and ocean fluxes. To make the direction clear, we reformulate the sentence in the introduction as: "The objective of this study is optimizing the natural $CO_2$ fluxes in order to provide a background for estimating the fossil $CO_2$ emissions where the advantage of high-resolution approach is more evident."

Replies to comments by Anonymous reviewer #2

RC2

The 'Technical note' by Maksyutov and co-authors describes the technical details of a CO2 flux inversion technique based on the coupled Eulerian-Lagrangian transport model NIES-TM-FLEXPART. The coupled system operates at a high spatial resolution of CO2 fluxes of 0.1° x 0.1° globally and also attempts a flux inversion at this resolution. As such the approach is novel and promising. The paper is well structured and written. The performance of the inversion is documented by time series comparisons/evaluations

of different data sources assimilated and not assimilated by the system and a comparison to another independent inversion system. However, the inverted flux fields are never shown/discussed in the manuscript, which makes it rather difficult to judge if the inversion yielded reasonable results. Even though the manuscript is a 'technical note', it would be very beneficial to overcome this shortcoming (detailed suggestion below) before publication.

Major comments

Section 5: Evaluating the performance of an inverse modelling system is not straightforward. Restricting this evaluation to a comparison of model skill for the assimilated and additional independent concentration time series is not sufficient. By over-fitting the flux fields a very good agreement of the posterior concentrations with the observations may be achieved but the flux fields may contain unrealistic detail in order to achieve this. Given the large degree of freedom in the fluxes, as indicated by the fact that grid and time resolved fluxes are inverted from a relatively limited set of observations, there seems to be a high risk for the presented inversion setup to over-fit the solution. Since no flux fields are presented, it is impossible to judge this possibility. Therefore, I would encourage the authors to extend their discussion of results in section 5 to include a brief analysis of the obtained flux fields. I can see that the authors have planed this for a later publication and, hence, I don't think this needs to be very quantitative here, but the presented flux fields should document the validity of the inversion approach. A qualitative comparison with flux fields obtained from CarbonTracker (as done for the concentration time series) would also be beneficial.

Reply: To illustrate the flux adjustments by inversion, we add the flux comparison figure for selected regions, comparing prior, posterior and CarbonTracker fluxes.

Minor comments

P3,L18: Resolution of coupled Eulerian-Lagrangian models. I find it a bit misleading to say that the transport in these models is run at a resolution of (as in the cited publication) 1 km. Yes, technically the transport is not run at any fixed resolution in the Lagrangian sense, but the driving meteorology is still determining what scales of motion can be correctly resolved by the model. The Lagrangian model may still have some skill in the sub-resolved range, but basically it degenerates to a Gaussian plume at these scales with constant wind direction, speed and dispersion characteristics. This fact is not sufficiently highlighted throughout the manuscript. Another example is P4, where the transport system for the current study is introduced. The driving meteorology is 1.25°x1.25° this is certainly not sufficient for a detailed transport description in complex, mountainous terrain, but also not for coastal areas. Since sites from both environments are contained in the list of sites used for assimilation, this limitation should be discussed in more detail. Next to spatial resolution also temporal resolution is important for a transport description in the mentioned environments. Although in the

following, the use of observations was restricted to certain times of the day, these observations still carry the transport history for a longer period and if temporal variability in the transport is not sufficiently described may lead to biases in the simulated concentrations as well.

Reply. We revised the sentence at P3, L18 and added a note on resolution in model description section.

P5,L13f: If I understand this correctly, there is no diurnal cycle of the biospheric CO2 flux considered in the model setup. How much is this simplification limiting the model performance? Not all sites used for the inversion are remote coastal sites but are surrounded by dense vegetation. How much does the constant diurnal flux and the restriction to afternoon observations introduce a bias in the flux inversion? In general, I have the feeling that the low temporal resolution of fluxes does not keep up with their high spatial resolution in the current setup.

Reply. Following the comment, we estimated the impact of using daily constant flux in place of hourly varying by running forward simulation with hourly and daily mean versions of SIB model fluxes (Denning et al. 1996), as used in Transcom intercomparison (Law et al. 2008), and added a Figure showing the simulated seasonal mean biases. It appears the hourly fluxes produce $0.5 - 1$ ppm lower simulated $CO_2$ in summer with respect to daily mean fluxes. Thus, it will be clearly helpful to apply diurnally varying fluxes in place of those with daily temporal resolution.

P6,L8: How much do the posteriori fluxes actually depend on the chosen biospheric flux climatology? Given the large year-to-year variability in biospheric fluxes, is it sufficient to operate with a climatology of prior fluxes? Was this evaluated by choosing a different averaging interval or even individual year for the prior climatology.

Reply. Use of climatology leads to degradation of prior simulation, but it allows for extending the simulations to years when the simulation is not available (we only had the data till 2010). For some regions, inverse corrections to prior are substantial, and look like exceeding the interannual variability. Added a note: "Although the use of climatology in place of original fluxes degrades the prior, the posterior fluxes show significant departures from the prior, thus reducing the impact of prior variations."

Section 3.3: Are all biomass burning emissions considered to be released at the model surface or was any kind of vertical emission profile used? Again, this may be crucial when considering transport simulations at the mesoscale.

Reply. Added a text 'The fluxes are input to the model at the surface, which may lead to underestimation of injection height for strong burning events and occasional overestimation of biomass burning signal simulated at surface stations.'

P7, L9 and P9, L21: According to the first text location only afternoon samples were

considered for the inversion. However, according to the second location daily average footprints of the Lagrangian model were applied. If that is really the case the footprints are not representative for the used observations, largely neglecting temporal variability once again!

Reply: To avoid confusion, removed 'daily average', added sentence: 'The flux footprints are saved at daily or hourly timestep, depending on available surface fluxes.'

P9, L25ff: The description of the forward model steps. From this description it is not clear to me how the concentration increments from the different models are added to avoid double counting of the fluxes (step 2.c). Shouldn't the Eulerian model use different fluxes than the Lagrangian (cropping those that are covered by the Lagrangian model)?

Reply: Added clarification: 'For each observation event, the fluxes used in Eulerian and Lagrangian components are separated by coupling time, so that there is no double counting of fluxes for the same date and time in the coupled model simulation.'

P10, L24: Initial conditions are used from an optimised run from the previous year. But then the question remains how the previous year was initialised. Was this done with a spin-up run?

Reply: Added: 'When the optimised fields are not available, the output of multiyear spin-up simulation is used, with same adjustment to South Pole observations.'

Section 5.1: Why is RMSE used as the sole estimator of model performance? RMSE will decrease even if only the baseline fits better after optimisation. Most of the regional flux information, however, is stored in the peak concentrations, for which a more robust performance estimator could be a bias corrected RMSE or the coefficient of determination. The bias should be reported as well. Taylor skill score could be another performance parameter that would be more suited to focus more on the short term variability.

Reply: We added a site bias data to the plot.

Sites: It would be useful to see the sites and the aircraft locations on a map. Would help to judge which areas are not well covered by assimilated observations in comparison to validation data. Such a plot should also contain the information of flask vs. continuous observations.

Reply: Site map added, with separate symbols for flask, continuous and aircraft observations.

Technical comments

P3,L10: Start a new sentence after '... Kaminski et al. (2001). This is addressed by ..."

Corrected

P12,L14: 'previous' instead of 'pervious'.

Corrected

Figure 1, caption: Label as 'Examples of ...'

Corrected

Figure 3+4: Since the x-axis is not along a continuous variable, I would suggest to not include lines in the plot or even use a barplot instead of symbols. The lines are just confusing and have no physical meaning

Reply: revised as suggested

References

Jones, L. A., Kimball, J. S., Reichle, R. H., Madani, N., Glassy, J., Ardizzone, J. V., Colliander, A., Cleverly, J., Desai, A. R., Eamus, D., Euskirchen, E. S., Hutley, L., Macfarlane, C., and Scott, R. L.: The SMAP Level 4 Carbon Product for Monitoring Ecosystem Land–Atmosphere $CO_2$ Exchange, IEEE Transactions on Geoscience and Remote Sensing, 55, 6517-6532, 10.1109/TGRS.2017.2729343, 2017.

Submitted in behalf of the authors by Shamil Maksyutov, Sep 15, 2020

---

## Author Response (AR2)

Authors comments on 'Technical note: A high-resolution inverse modelling technique for estimating surface  $CO_2$  fluxes based on the NIES-TM – FLEXPART coupled transport model and its adjoint', submitted by S. Maksyutov in behalf of authors.

Authors are grateful to reviewers and editor for checking a revised manuscript and providing valuable suggestions and comments. We made effort to revise manuscript reflecting the reviewers' comments to the extent possible within a given time frame and hope the revisions and clarifications help improving the manuscript and clear the doubts on merits of publication.

In the following text, the reviewer's comments are shown in italics, replies as normal text and revisions to the manuscript are shown in color.

**Comments by reviewer #1**

The paper presents the inverse modeling system developed at NIES, based on the NIES-TM and FLEXPART transport models. The aim of the paper, as I understand it, is to present the system, as well as introduce some technical and developments, and in particular a new method to compute flux covariance matrices. The stated aim of the development is eventually to go towards optimizing anthropogenic CO2 emissions at a high resolution.

The authors have addressed several of the comments from the initial round of review, and the paper has overall improved. However, I remain unconvinced that it should be published in ACP: The main unique features of the inverse model are the construction of flux covariance matrices using implicit diffusion instead of classical Gaussian correlation functions, and the high-resolution transport achieved via a coupling between NIES-TM and FLEXPART. I think that both are good ideas, which deserve to be explored and eventually published. But they are also very technical, rather out of the scope of ACP (it would fit better in a journal like GMD). A publication in ACP could still be justified if it was demonstrated that these developments lead to considerable changes in our comprehension of CO2 fluxes, or if they enabled inversions that would not otherwise be possible. Unfortunately, neither is done in the paper:

Reply: We consider our study to be in second category, ("enabling inversions that would not otherwise be possible") that is advancement towards using high resolution transport in the global inversion, which is computationally prohibitive when applying a conventional Eulerian model.

Although GMD looks as a good alternative for publication, it has some inclination towards publishing on modelling software, rather than modelling techniques. We leave it to editors to decide if the scope of ACP Technical note can accommodate the presented content.

The merit of the proposed approach is based of computational advantage the coupled model has over Eulerian model. We added this note to the Section 2 as new text

"Coupled transport model is more computationally efficient in comparison to the Eulerian model operating at same resolution. It was confirmed by Belikov et al, (2016) that coupled model with Lagrangian model run at resolution of  $1^{\circ}\times1^{\circ}$  performs similarly when using together with Eulerian model at either  $1.25^{\circ}\times1.25^{\circ}$  or  $2.5^{\circ}\times2.5^{\circ}$  resolution, and only can see performance degradation when using  $10^{\circ}\times10^{\circ}$  resolution Eulerian model."

Another outcome of the study is demonstration of using the coupled model and its adjoint in global flux inversion based on a variational optimization, which has not been attempted before for  $CO_2$  tracer, especially with effective transport model resolution of 0.1° at global scale.

We also obtained preliminary results on ability to reproduce the observed concentrations. Unfortunately, we do not see a significant improvement over existing systems such as CarbonTracker in most difficult cases such as resolving  $CO_2$  variability over North American continent in summertime, which is an indication of a need to use higher resolution meteorology and more accurate prior fluxes. Still this negative result can be useful as guidance for future improvements.

- There is no real really focus on the CO2 fluxes. Compared to the initial submission, they added a section presenting some optimized fluxes, but this is clearly just illustrative, the authors themselves state that "more detailed comparisons [...] should be made after improving the inversion setup". I understand that choice of not focusing on the fluxes during the development phase of the system, but it then it only leaves the quality and novelty of the inversion setup as argument for publication.

**Reply:**

We are in process of preparing more accurate wind data and diurnally varying prior fluxes, which are necessary steps for reducing biases and completion of a production system, however the algorithms, core components for the transport and inverse model will not be changed, thus present description will be relevant. As this study provides a first description of the inverse modeling technique based on global coupled transport and variational optimization, implemented as working system, we consider there is sufficient novelty and significance to publish it.

- On that second point, the authors unfortunately chose a setup that doesn't put their model at an advantage compared to existing inverse models: the approach is supposed to facilitate the optimization of CO2 fluxes at a very high resolution, but they used correlation distances of 500 km, which, concretely, means that adjacent pixels are not resolved independently at all by the inversion, and the fluxes are only adjusted by large patches of a few hundred kms (as illustrated in Fig 5, left). What is then the advantage, in terms of scientific results and in terms of computational efficiency, compared to an inversion that would optimize fluxes only at a resolution of just 1°? Would their system still perform well in conditions where others would struggle? If the correlation distances were set small enough so that the inversion can adjust some of the fine-scale spatial structure of the fluxes, would the system still perform well (intuitively, I imagine that the number of iterations required will increase exponentially when the correlation distances are reduced ...)? Maybe I am excessively pessimistic, but I think it should be relatively easy for the authors to prove me wrong!

**Reply:**

Short survey of regional studies on "high-resolution inversion" reveals that the gap between grid size and the correlation distances is not unusual. A number of studies on high-resolution inversion (He et al. 2018, Schuh et al, 2010, Villalobos et al 2020, and this study) run into a risk of misunderstanding about the effective resolution of inversion, which in fact turns out to be restricted by available observations and regularization. Same applies to the inversions made in moderate resolution and working at model grid scale. Practically 'high resolution inversion' only means the optimization on fine grid scale decided by resolution of transport model, leading to a large dimension of the control space, and that dimension is not connected directly to the number of degrees of freedom the observations can effectively constrain.

The merit of using shorter correlation length have been considered in regional inversions that operate at grid size of 40-80 km. Still the correlation lengths of 500 km to 1000 km are used most widely in regional settings and are considered as proper balance between available information content and desired detail of regional fluxes. For example, in regional inversion over continental USA made with model grid size of 40 km by Schuh et al, (2010), or over Australia with 80 km grid size (Villalobos et al, 2020). We refer to study by Schuh et al, (2010) to diffuse impression that the choice of correlation length is arbitrary or excessively conservative, by adding a sentence "The rationale of applying correlation distance of 500 km in case of regional inversion over continental USA with model grid size of 40 km was discussed by Schuh et al, (2010)." to section 4.2.

- The use of low-resolution meteorology is understandable for a model development paper, but it makes it even more complicated to judge the interest of the work presented. Furthermore, the authors use a comparison to CarbonTracker as a sort of benchmark, but there are so many differences between the two systems that it's hard to retain anything from that comparison: the transport models are different, the meteo data driving them is different, the inversion scheme is different, the prior constraints are different, if anything, it shows the NIES-TM-FLEXPART system in an un-necessarily unfavourable view.

**Reply:**

CarbonTracker shows a favorable performance in terms of posterior fit to observations, even when compared to higher resolution models (Zheng et al., 2020), which makes it a good reference for comparison. We are developing a new tracer transport and inverse modeling technology and face a choice of maintaining the older one in parallel with developing a new one, and while there is a need to improve our system in many aspects, achieving a similar performance to reference systems such as CarbonTracker, even

without outperforming those, provides a rationale for a practical decision to adopt a new technology as we not likely to lose by doing that.

My recommendation in the initial submission oscillated between rejection (with some encouragement to improve and resubmit the manuscript to another journal, maybe GMD), and a major revision. This was based pretty much on the same reasons as listed above, and since the authors only offered a minor revision and did not resolve these issues, I cannot change my recommendation.

If the authors want to persist with publishing this paper in ACP, then I would recommend either enriching the paper with a comprehensive set of sensitivity inversions, to assess the use of each of the unique features of their setup, or alternatively, produce a "reference" inversion, using their best possible setup (I.e. including the needed improvements listed in the last sentence of Section 5.2).

**Reply:**

We agree that those improvements are necessary before finalizing a system taking full advantage of the presented technique for applications for carbon cycle studies. But at this stage we only report completion of an algorithm development, achieving a new capability and summarize results of the tests with real data that give direction for further improvements.

**Minor comments**

Besides the major comments above, the paper is overall in a very good shape (and given my overall review, I did not spend too much time in looking for very minor issues). One issue though is the way FLEXPART footprints are computed: depending on the release time, the footprints will span a period of 2 to 3 days. I don't know what the consequences are in practice, but I guess that it can introduce some form of systematic bias in the daily cycle, since the length of the footprints will be a function of their longitude and release time. Since the footprints ingested by the system are eventually only 2D, this may not matter too much, but this should be verified and justified in the paper.

**Reply:**

The possibility of uneven treatment of observations made at different time zones is mentioned correctly. It could be avoided by coupling to Eulerian model at higher frequency (*eg* every hour). Still we consider it a minor effect as both models, Lagrangian and Eulerian, simulate diurnal cycle of vertical mixing and diurnal variability of fluxes. It is also a related to a problem of choosing the optimal time length of Lagrangian transport (3 or 5 or 7 days) which is decided by a balance between accuracy and a need to hold all 2-D responses for the assimilation window in memory between iterations in the current version of the model. The correctness of the treatment of fluxes a coupling time boundary is discussed in the section 4.1.

Comments by reviewer #2

This is my second review of the manuscript. My previous comments were answered sufficiently and my suggestions for further clarification and additional results were followed. Hence, I only have a few minor remarks at this point and would suggest the manuscript for publication after these points were addressed.

Section 5.1 and Figs. 3 + 4: I had asked for additional performance parameters to complement the discussion on RMSE. In addition to the previous version, biases (posterior I suppose? not mentioned in the figure caption) are shown now for the assimilated data but not for the validation sites (Fig 4). There is also not mention of these biases in the text.

Reply: The biases for validation site are also added to the plots and mean statistics presented in the text.

The posterior bias by itself is not sufficient to understand the improvements achieved by the inversion. I would still suggest that bias, bias corrected RMSE and correlation coefficients should be discussed together to understand where and how the improvements were gained (baseline signal as laid down in bias, or regional signal as seen in the correlation and/or bias corrected RMSE). This discussion should be done for both assimilated and validation data in the same way and showing the same parameters. This could be done in a more summarizing way as right now where it is difficult to identify individual changes.

We have added the posterior biases to the validation plots on Fig 4. In the discussion, we add description on  $r^2$  increase by optimization to the text (rather than adding to individual site plots), as it was rightfully noticed that the amount of detail on the Figure can become excessive.

Section 5.2 and Fig 5: I am not sure that I understand which fluxes are shown in Fig 5. Does "optimized correction" refer to the difference between posterior and prior? And is this difference only for the biospheric flux? I am also missing a short comment on the shown maps. For example a short discussion and interpretation of the observed changes. Is there any reason why largest changes are assigned to Russian coast? Is this climatologically feasible?

Reply: We added explanation that flux correction shown for one month, and is not representative of a seasonal or annual mean, and is made as combination of prior flux shape and scaling factor, with later likely to have a dip between continent and Japanese islands.

Fig 6 illustrates nicely that the system results in comparable results as compared to CT (which obviously does not prove anything since CT may also be biased). However, I wonder why only 6 of the TRANSCOM regions are displayed. What happened in the tropical regions?

Reply: To clarify, we added to the text that the fluxes do not include fossil fuel emissions (not optimized). The idea behind selecting the regions was to show examples of regions well constrained by the observations and weakly constrained. Some tropical regions were not shown as the fluxes there are not well constrained thus posterior fluxes appear to be similar to prior. In the revised version we added plots for two more tropical regions to Figure 6.

Revisions in response to comments by Reviewer #2

Section 5.1

**Revised:**

[revised manuscript text omitted]

**Other revisions to the text**

Section 4.2 Revised: Reference to Miller et al. (2020) (was 2019, paper changed from discussion to final)

Figure A3 Mistype in right axis title corrected after editor's suggestion.

References Added reference to Schuh et al., (2010)

[revised manuscript text omitted]

---

## Author Response (AR3)

Dear editors

We received following suggestion to introduce needed correction to manuscript:

*Before publishing, however, the manuscript should be carefully read and corrected for spelling and grammar. Specifically, there are many missing indefinite articles („a" or „an") as well as missing definite articles („the") that need to be added. An example is in line 15 „The use of high-resolution atmospheric transport in global CO2 flux inversion", were „a" should be in front of „inversion", or it should be „inversions". As there are quite a number of such missing articles in the manuscript, may be one of the native english speaking coauthors can help with this.*

We are sorry for failing to make a better effort on the earlier stage. The manuscript has been checked by coauthors and needed improvements are made as highlighted in following manuscript. We made corrections whenever we find problems with grammar, use of articles, punctuation, plural/single forms, orthography and missing words (without altering the logic and meaning of the text). We are grateful to editorial team for kind assistance with our manuscript.

In behalf of authors

Shamil Maksyutov, Rajesh Janardanan

[revised manuscript text omitted]